# Spectral Analysis of Molecular Kernels: When Richer Features Do Not Guarantee Better Generalization

## Abstract

Understanding the spectral properties of kernels offers a principled perspective on generalization and representation quality. While deep models achieve state-of-the-art accuracy in molecular property prediction, kernel methods remain widely used for their robustness in low-data regimes and transparent theoretical grounding. Despite extensive studies of kernel spectra in machine learning, systematic spectral analyses of molecular kernels are scarce. In this work, we provide the first comprehensive spectral analysis of kernel ridge regression on the QM9 dataset, molecular fingerprint, pretrained transformer-based, global and local 3D representations across seven molecular properties. Surprisingly, richer spectral features, measured by four different spectral metrics, do not consistently improve accuracy. Pearson correlation tests further reveal that for transformer-based and local 3D representations, spectral richness can even have a negative correlation with performance. We also implement truncated kernels to probe the relationship between spectrum and predictive performance: in many kernels, retaining only the top 2% of eigenvalues recovers nearly all performance, indicating that the leading eigenvalues capture the most informative features. Our results challenge the common heuristic that "richer spectra yield better generalization" and highlight nuanced relationships between representation, kernel features, and predictive performance. Beyond molecular property prediction, these findings inform how kernel and self-supervised learning methods are evaluated in data-limited scientific and real-world tasks.

## 1 Introduction

Accurate molecular property prediction lies at the heart of modern molecular and materials-discovery pipelines, where rapid estimation of properties can accelerate screening, design, and optimization Bohacek et al. (1996); Reymond (2015); Goh et al. (2017); Kailkhura et al. (2019); Shen & Nicolaou (2019); Schapin et al. (2023); Kuang et al. (2024). In molecular property prediction, two major modeling paradigms have emerged: (i) neural network–based and (ii) kernel-based models. Neural networks (NN) have advanced rapidly, driven by large datasets and architectures tailored to molecules, such as graph neural networks (GNN) and equivariant NNs Jiang et al. (2021); Le et al. (2022); Ju et al. (2023). In contrast, traditional kernel methods excel in low-data regimes, offering strong generalization capabilities that make them especially valuable for sample-efficient tasks such as active learning and Bayesian optimization in material discovery Griffiths et al. (2023); Ralaivola et al. (2005); Bartók et al. (2013); Khan et al. (2023). Their non-parametric nature enables them to capture complex similarity structures without requiring extensive hyperparameter tuning or massive training datasets. Kernel methods also underpin some of the most successful machine-learned interatomic potentials Kamath et al. (2018); Thant et al. (2025), enabling accurate predictions of atomic forces and energies across diverse chemical systems.

Unlike NNs, which adapt features from data, kernel methods rely on fixed kernels tailored to specific representations Rasmussen & Williams (2005). The design of effective molecular kernels is particularly challenging: molecules may be represented using Cartesian or internal coordinates, cheminformatics descriptors such as Morgan fingerprints, or graphs of atoms and bonds Griffiths

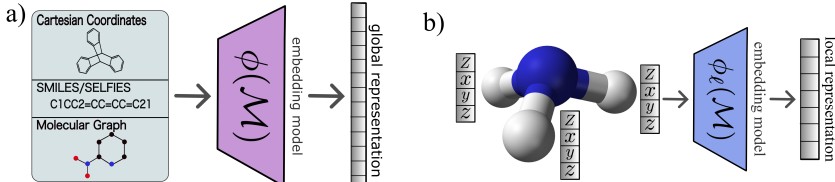

Figure 1: Molecular representation generation workflow compatible with kernel functions; (a) global ($\phi(\mathcal{M})$) and (b) local ($\phi_\ell(\mathcal{M})$) molecular representations, where $\mathcal{M}$ represents a molecule.

et al. (2023). Each choice defines a different notion of similarity, with graph kernels in particular motivated by the notoriously hard graph isomorphism problem. More recently, pretrained molecular embedding models based on GNNs or transformers have emerged as alternative molecular representations Praski et al. (2025).

Traditionally, the quality of different molecular kernels is evaluated primarily by their *test-set performance in particular downstream tasks*. While informative, this evaluation overlooks deeper questions:

*How well does a kernel capture the structure of the target function?*
*What does this reveal about the quality of molecular representations for downstream tasks?*

Interestingly, machine learning theorists have also asked the same questions and answered with a keyword: *kernel spectrum*. In recent years, due to the theory of the neural tangent kernel in overparameterized neural networks Jacot et al. (2018), machine learning theorists have reignited interest in the performance guarantee of kernel methods Arora et al. (2019), especially the ones that depend on the kernel spectrum Mallinar et al. (2022); Li et al. (2023); Barzilai & Shamir (2024); Cheng et al. (2024b). In parallel with advances in kernel theory, the field of self-supervised learning (SSL) has already implemented model evaluation depending on the feature spectrum of the SSL model with label-less data Agrawal et al. (2022); Garrido et al. (2023), with the heuristic to choose the model with the richest feature spectrum and the belief that "richer features yield better generalization".

**Contribution** In this work, we investigate whether the aforementioned theoretical insights are applicable in the context of molecular chemistry. Our key contributions are:

- **Comprehensive spectral analysis of molecular kernels.** We present the first systematic spectral analysis of molecular kernels for molecular property prediction on the QM9 dataset, encompassing three global, three local, and three transformer-based encodings, as well as extended connectivity fingerprint (ECFP) kernels. To our knowledge, we are also the first to apply kernel ridge regression on pretrained transformer-derived features with various kernels, achieving improved performance over the commonly used linear regression baseline.

- **Correlation analysis.** We compute four spectral metrics that quantify feature richness and examine their relationship with the average $R^2$ score. Pearson correlation tests reveal that richer features do not universally yield better performance; notably, for transformer-based and local 3D representations, all spectral metrics even indicate a negative correlation, challenging common assumptions in kernel theory and self-supervised learning (SSL).

- **Implementation of truncated kernels.** We extend the concept of *truncated kernels* from Amini et al. (2022) to ECFP-based kernels, quantifying the fraction of eigenvalues required to recover 95% and 99% of the original performance. Our results show that the top eigenvalues capture most of the important features, further questioning the general belief that richer spectra necessarily improve generalization.

**Organization** The paper is structured as follows. In Section 2, we review the relevant background and key concepts. Section 3 presents our experimental methodology and results. In Section 4, we discuss the novelty, limitations, and potential future directions of our work. Due to space constraints, additional experimental results, analyses, and discussions are provided in the Appendix.

## 2 BACKGROUND

In molecular property prediction, the inputs are molecules $\mathcal{M}$, discrete objects without an inherent Euclidean representation. This necessitates the use of domain-specific representations, each inducing a corresponding kernel that encodes molecular similarity.

**Molecular Representation** Molecular representations for kernel methods are typically grouped into two categories: (i) *global descriptors*, which encode information about the entire molecule, and (ii) *local descriptors*, which capture the environments surrounding individual atoms; see Fig. 1 for illustration. Beyond handcrafted descriptors, representation learning approaches, ranging from autoencoders to natural language processing architectures, have also been developed in semi-supervised and unsupervised settings, mapping string inputs into high-dimensional feature vectors Praski et al. (2025). In this work, we refer to **3D** kernels as those, either local or global, that are based on Cartesian coordinates. For more details, please refer to Section A.

**Kernel Ridge Regression** A molecular kernel $k$ maps any two molecules $\mathcal{M}_i$, and $\mathcal{M}_j$ into a real number, and the kernel matrix $\boldsymbol{K} \in \mathbb{R}^{n \times n}$ with entries $\boldsymbol{K}_{ij} = k(\mathcal{M}_i, \mathcal{M}_j)$ is symmetric and positive semi-definite. Note that such a kernel $k$ is associated with a reproducing kernel Hilbert space (RKHS) $\mathcal{H} = \{\sum_i a_i k(\mathcal{M}_i, \cdot) : a_i \in \mathbb{R}\}$ where its dot product is defined by the kernel: $\langle k(\mathcal{M}_i, \cdot), k(\mathcal{M}_j, \cdot) \rangle_{\mathcal{H}} = k(\mathcal{M}_i, \mathcal{M}_j)$. Now, we consider molecular property prediction: given a training set of molecules with prediction objectives $\{(\mathcal{M}_i, y_i)\}_{i=1}^n \subset \mathcal{M} \times \mathbb{R}$, the predictor $\hat{f}$ can be obtained via kernel ridge regression (KRR):

$$\hat{f}(\mathcal{M}) = \sum_{i=1}^n \alpha_i k(\mathcal{M}_i, \mathcal{M}), \tag{1}$$

where $\alpha_i = [(\boldsymbol{K} + \lambda \boldsymbol{I})^{-1} \boldsymbol{y}]_i \in \mathbb{R}$, $\boldsymbol{K} = [k(\mathcal{M}_i, \mathcal{M}_j)]_{i,j}^n \in \mathbb{R}^{n \times n}$, $\boldsymbol{y} = [y_0, \cdots, y_n] \in \mathbb{R}^n$, and $\lambda \geq 0$ is the regularization constant. Kernel ridgeless regression is a special case where $\lambda = 0$, in such a case, the kernel could overfit the training data, benign, tempered, or catastrophic Mallinar et al. (2022).

**Truncated Kernel Ridge Regression** Since the RKHS $\mathcal{H}$ is typically infinite-dimensional, and KRR inherently biases toward eigenfunctions associated with larger eigenvalues Basri et al. (2020), it is natural to consider truncating the kernel spectrum by retaining only the top eigen-components. Recently, this idea has been brought to supervised learning settings in the form of truncated kernel ridge regression (TKRR) Amini et al. (2022). Formally, fix a training set $\{(\mathcal{M}_i, y_i)\}_{i=1}^n$, a truncation level $r \leq n$, there exists a kernel $k^{(r)}$ such that its kernel matrix $\boldsymbol{K}^{(r)}$ is the rank-$r$ approximation of the original kernel matrix $\boldsymbol{K}$. In other words, if the original kernel matrix admits the eigen-decomposition $\mathbf{K} = \sum_{k=1}^n \mu_k \mathbf{u}_k \mathbf{u}_k^\top$, then the truncated kernel matrix is equal to

$$\mathbf{K}^{(r)} = \sum_{k=1}^r \mu_k \mathbf{u}_k \mathbf{u}_k^\top. \tag{2}$$

To obtain the TKRR predictor, we need to replace the kernel by its truncated version $k^{(r)}$ in Eq. 1. From the computation perspective, the kernel matrix is readily given by $\boldsymbol{K}^{(r)} = \sum_{k=1}^r \mu_k \mathbf{u}_k \mathbf{u}_k^\top$ as in Eq. 2. However, its value $k^{(r)}(\mathcal{M}_i, \mathcal{M})$ on any new test point $\mathcal{M}$ is empirically intractable. To overcome this hurdle, we introduce the approximated truncated kernel ($\tilde{k}^{(r)}$),

$$\tilde{k}^{(r)}(\mathcal{M}_i, \mathcal{M}) = [\mathbf{U}_{\leq r} \mathbf{U}_{\leq r}^\top \mathbf{k}]_i,$$

where $\mathbf{U}_{\leq r} = (\mathbf{u}_k^\top)_{k=1}^r \in \mathbb{R}^{n \times r}$, $\mathbf{k} = (k(\mathcal{M}_j, \mathcal{M}))_{j=1}^n \in \mathbb{R}^n$. Please refer to Section D for the properties of $\tilde{k}^{(r)}$.

**Self-Supervised Learning** The heuristic to choose the model with the richest feature spectrum and the belief that "richer features yield better generalization" impacts the evaluation of model quality in the SSL context Agrawal et al. (2022); Garrido et al. (2023). In the kernel method, instead of the spectrum of the covariance matrix of the features, one studies the empirical kernel spectrum,

that is, the eigenvalues of the kernel matrices, and yields similar conclusions Mallinar et al. (2022); Cheng et al. (2024a). Intuitively, richer features mean the feature vectors span into different directions in the ambient space, capturing as many possible details from the input. Formally, given a spectrum $\{\mu_1, \mu_2..., \mu_p\}$, $p \in \mathbb{N} \cup \{\infty\}$ in decreasing order, Agrawal et al. (2022); Mallinar et al. (2022) assume that the spectrum follow a power law: $\mu_j \propto j^{-\alpha}$ for some $\alpha > 0$, which can be computed empirically from linear regression on the log-spectrum $(\log \mu_j)_j$ and its index $j$. Smaller $\alpha$ indicates richer features. Alternatively, Huh et al. (2023); Garrido et al. (2023) proposed using spectral Shannon entropy (SSE) to measure feature richness; a higher SSE indicates richer features. Other metrics like intrinsic dimension (ID) and stable rank (SR) are also commonly used in spectral analysis Ipsen & Saibaba (2024). For detailed definitions, please refer to Section C.

## 3 RESULT

In this paper, we evaluate kernel ridge regression for molecular kernels on the QM9 dataset Ramakrishnan et al. (2014), a benchmark of $\sim$134,000 small organic molecules containing up to nine heavy atoms (C, O, N, F). The molecular properties in QM9 were computed using density functional theory at the B3LYP/6-31G(2df,p) level. Our experiments focus on predicting the HOMO–LUMO gap (Gap), internal energy at 0 K ($U_0$) and 298.15 K ($U_{298}$), heat capacity ($C_V$), enthalpy ($\Delta H$), Gibbs free energy ($G$) at 298.15 K, and zero-point vibrational energy (ZPVE).

**Molecular Representations**  We evaluate kernels constructed from four distinct categories of molecular representations.

1. **Fingerprint-based:** We use multiple kernels that rely on the ECFPs global representation, e.g., Tanimoto, Dice, Otsuka, Sogenfrei, Braun-Blanquet, Faith, Forbes, Inner-Product, Intersection, Min-Max, and Rand kernels. Kernels' details in Section A.1.1.

2. **Pretrained transformer-based:** We extract features from pretrained molecular transformers (SELFIESTED, SELFormer, and MLT-BERT) with string-like input of molecules like SELFIES, and build Gaussian, Laplacian, and linear kernels on top of the features; more details in Section A.1.2.

3. **Global 3D descriptors:** We employ representations that capture entire molecular geometry, such as Coulomb matrix (CM), bag of bonds (BOB), and SLATM, and build isotropic Gaussian, Laplacian, and linear kernels on top of these representations; more details in Section A.1.3.

4. **Local 3D descriptors:** We consider local structural descriptors that encode pairwise atomic environments, including local SOAP Bartók et al. (2013) and related local descriptors such as FCHL19 Christensen et al. (2020) and ACSF Behler (2011). A special note is that linear kernels are not well-defined for local 3D descriptors, as these kernels are inherently designed to compare molecules through pairwise local environment similarities for similar atoms rather than global vector embeddings; more details in Section A.2.

In particular, ECFPs were generated with `RDKit` (radius = 3, vector size = 2048), while FCHL19, SLATM, and ACSF descriptors were computed using the `QMLcode` library Christensen et al. (2017). SOAP representations were obtained with the `DScribe` package Himanen et al. (2020); Laakso et al. (2023), using default Gaussian-type radial basis functions.

**Hyperparameter Choice**  The hyperparameters associated with the molecular representations were kept fixed, as per Khan et al. (2023), to ensure consistency in the representations. For Gaussian and Laplacian kernels, the length scale parameter $\sigma_\ell$ was restricted to values of $10^2$ or $10^4$ before any training result. The regularization hyperparameter $\lambda$ is tuned separately for each representation through grid search combined with 5-fold cross-validation. The best configuration was then selected based on the highest $R^2$ score on the validation set. Results presented in Table 1 correspond to test sets of $10,000$ molecules, with all models trained on $5,000$ randomly selected molecules.

**Spectral Metrics**  Given a kernel matrix $\boldsymbol{K}$, we compute its empirical eigenspectrum $\mu_1, \ldots, \mu_n$ and evaluate four spectral metrics to quantify its richness: polynomial decay rate ($\alpha \downarrow$), spectral Shannon entropy (SSE $\uparrow$), intrinsic dimension (ID $\uparrow$), and stable rank (SR $\uparrow$). The arrows indicate the direction corresponding to richer spectral features, with formal definitions provided in Section C.

In our experiments, we find that most kernel spectra are dominated by a single large leading eigenvalue ($\mu_1$), followed by a sharply decaying tail (see Fig. 3 and figures in Section B.1). To more accurately capture spectral richness, we also compute these four spectral metrics on truncated spectra by removing the top three eigenvalues and restricting to the top 50% of the spectrum. These truncated values are reported in parentheses in Table 1.

Table 1: Comparison of spectral metrics and $R^2$ scores for kernel regression with different molecular representations. The $R^2$ scores are computed on a test set of 10,000 random molecules (maximum value 1; higher is better). The highest and second-highest averages for each molecular representation type are shown in **bold** and underline, respectively. The four spectral metrics quantify the richness of the kernel spectrum (direction indicated by arrows). Values in parentheses correspond to truncated spectra, used to mitigate the effect of outliers. The symbols indicate, $\dagger : \sigma_\ell = 100$, and $\ddagger : \sigma_\ell = 10^4$.

| Mol. Rep. | Kernel | $\alpha \downarrow$ | SSE ↑ | ID ↑ | SR ↑ | $R^2$ ↑ | | | | | | | |
|---|---|---|---|---|---|---|---|---|---|---|---|---|---|
| | | | | | | Gap | $C_V$ | $\Delta H$ | $U_0$ | $U_{298}$ | $G$ | ZPVE | Avg |
| ECFPs | Tanimoto | 0.7 (0.6) | 1693.5 (1363.6) | 13.7 (62.8) | 1.3 (8.2) | 0.826 | 0.752 | 0.719 | 0.719 | 0.719 | 0.719 | 0.861 | **0.760** |
| | Dice | 0.9 (0.8) | 429.9 (798.2) | 7.6 (42.2) | 1.2 (7.1) | 0.778 | 0.729 | 0.690 | 0.690 | 0.6901 | 0.690 | 0.842 | 0.733 |
| | Otsuka | 0.9 (0.8) | 427.1 (794.7) | 7.6 (42.3) | 1.2 (7.1) | 0.773 | 0.712 | 0.664 | 0.664 | 0.664 | 0.664 | 0.836 | 0.711 |
| | Sogenfrei | **0.5** (0.5) | **3110.5** (1851.8) | **40.5** (118.6) | **2.1** (13.8) | 0.844 | 0.722 | 0.669 | 0.669 | 0.669 | 0.669 | 0.855 | 0.727 |
| | Braun-Blanquet | 0.9 (0.8) | 423.5 (789.0) | 7.5 (42.5) | 1.2 (7.1) | 0.756 | 0.666 | 0.544 | 0.544 | 0.544 | 0.544 | 0.820 | 0.631 |
| | Faith | 0.9 (0.8) | 1.2 (782.8) | 1.0 (41.9) | 1.0 (7.0) | 0.765 | 0.703 | 0.638 | 0.638 | 0.638 | 0.638 | 0.828 | 0.692 |
| | Forbes | 0.9 (0.8) | 429.9 (798.2) | 7.6 (42.2) | 1.2 (7.1) | 0.739 | 0.702 | 0.666 | 0.666 | 0.666 | 0.666 | 0.826 | 0.704 |
| | Inner-Product | 0.9 (0.8) | 423.5 (789.0) | 7.5 (42.5) | 1.2 (7.1) | 0.756 | 0.666 | 0.544 | 0.544 | 0.544 | 0.544 | 0.820 | 0.637 |
| | Intersection | 0.9 (0.8) | 1.1 (782.8) | 1.0 (41.9) | 1.0 (7.0) | 0.764 | 0.703 | 0.638 | 0.638 | 0.638 | 0.638 | 0.828 | 0.692 |
| | Min-Max | 0.7 (0.6) | 1693.5 (1363.6) | 13.7 (62.8) | 1.3 (8.2) | 0.826 | 0.752 | 0.719 | 0.719 | 0.719 | 0.719 | 0.861 | **0.760** |
| | Rand | 0.9 (0.8) | 1.1 (782.8) | 1.0 (41.9) | 1.0 (7.0) | 0.765 | 0.703 | 0.638 | 0.638 | 0.638 | 0.638 | 0.828 | 0.692 |
| SELFIESTED | Gaussian † | 3.0 (2.9) | 1.0 (54.0) | 1.0 (7.5) | **1.0** (2.7) | 0.849 | 0.981 | 0.993 | 0.993 | 0.993 | 0.993 | 0.995 | **0.971** |
| | Laplacian ‡ | **0.9** (0.5) | 1.3 (1436.7) | 1.0 (47.6) | **1.0** (5.0) | 0.813 | 0.968 | 0.975 | 0.975 | 0.975 | 0.975 | 0.986 | 0.952 |
| | Linear | 2.0 (10.0) | 4.6 (55.8) | **1.4** (7.8) | **1.0** (2.7) | 0.817 | 0.971 | 0.981 | 0.981 | 0.981 | 0.981 | 0.988 | 0.957 |
| SELFormer | Gaussian † | 2.7 (2.6) | 1.1 (46.5) | 1.0 (6.9) | **1.0** (2.3) | 0.827 | 0.851 | 0.827 | 0.827 | 0.827 | 0.827 | 0.933 | 0.846 |
| | Laplacian ‡ | **0.9** (0.5) | 1.3 (1672.7) | 1.0 (57.3) | **1.0** (5.1) | 0.773 | 0.742 | 0.687 | 0.687 | 0.687 | 0.687 | 0.871 | 0.733 |
| | Linear | 8.3 (9.9) | **4.8** (45.3) | **1.4** (6.8) | **1.0** (2.3) | 0.805 | 0.817 | 0.779 | 0.779 | 0.779 | 0.779 | 0.915 | 0.808 |
| MLT-BERT | Gaussian | 4.0 (1.8) | 1.0 (16.9) | 1.0 (4.0) | **1.0** (1.9) | 0.757 | 0.855 | 0.938 | 0.938 | 0.938 | 0.938 | 0.891 | 0.894 |
| | Laplacian † | 1.1 (1.0) | 4.7 (252.0) | 1.3 (10.1) | **1.0** (2.5) | 0.675 | 0.818 | 0.841 | 0.841 | 0.841 | 0.841 | 0.883 | 0.820 |
| | Linear † | 9.3 (5.3) | 1.9 (16.0) | 1.1 (4.0) | **1.0** (1.9) | 0.682 | 0.826 | 0.859 | 0.859 | 0.859 | 0.859 | 0.871 | 0.831 |
| CM | Gaussian † | 1.7 (1.7) | 4.9 (103.9) | 1.3 (17.5) | 1.0 (6.4) | 0.598 | 0.967 | 0.997 | 0.997 | 0.997 | 0.997 | 0.997 | 0.936 |
| | Laplacian ‡ | 1.5 (1.5) | 1.6 (275.0) | 1.1 (27.6) | 1.0 (6.9) | 0.779 | 0.987 | 0.997 | 0.997 | 0.997 | 0.997 | 0.999 | 0.965 |
| | Linear | 9.2 (8.2) | 1.8 (47.3) | 1.1 (15.1) | 1.0 (6.6) | 0.439 | 0.905 | 0.998 | 0.998 | 0.998 | 0.998 | 0.995 | 0.904 |
| BOB | Gaussian † | 2.6 (2.5) | **9.0** (30.5) | **2.1** (6.4) | **1.1** (3.0) | 0.783 | 0.966 | 0.997 | 0.997 | 0.997 | 0.997 | 0.999 | 0.962 |
| | Laplacian ‡ | 1.5 (0.8) | 1.6 (917.2) | 1.1 (32.4) | 1.0 (4.2) | 0.891 | 0.994 | 0.996 | 0.996 | 0.996 | 0.996 | 1.000 | 0.981 |
| | Linear | 10.4 (9.4) | 3.1 (16.4) | 1.4 (5.0) | 1.0 (2.7) | 0.605 | 0.943 | 0.998 | 0.998 | 0.998 | 0.998 | 0.999 | 0.934 |
| SLATM | Gaussian† | 2.9 (2.8) | 1.2 (19.8) | 1.0 (5.3) | 1.0 (2.5) | 0.941 | 0.998 | 0.999 | 0.999 | 0.999 | 0.999 | 1.000 | **0.991** |
| | Laplacian ‡ | **1.3** (0.9) | 1.2 (687.5) | 1.0 (30.9) | 1.0 (5.7) | 0.934 | 0.996 | 0.996 | 0.996 | 0.996 | 0.996 | 0.999 | 0.988 |
| | Linear | 4.5 (4.4) | 1.8 (15.7) | 1.1 (4.7) | 1.0 (2.3) | 0.809 | 0.995 | 0.999 | 0.999 | 0.999 | 0.999 | 1.000 | 0.971 |
| SOAP | Gaussian † | 2.5 (0.1) | 1.0 (1737.8) | 1.0 (24.2) | 1.0 (1.9) | 0.789 | 0.991 | 0.998 | 0.998 | 0.998 | 0.998 | 1.000 | 0.967 |
| | Laplacian † | 1.5 (1.4) | 1.0 (46.0) | 1.0 (3.8) | 1.0 (1.4) | 0.865 | 0.997 | 0.998 | 0.998 | 0.998 | 0.998 | 1.000 | 0.979 |
| FCHL19 | Gaussian † | 4.3 (2.1) | 1.0 (7.2) | 1.0 (2.2) | 1.0 (1.3) | 0.876 | 0.997 | 0.997 | 0.997 | 0.997 | 0.997 | 1.000 | 0.980 |
| | Laplacian † | 1.5 (1.4) | 1.1 (52.0) | 1.0 (4.9) | 1.0 (1.9) | 0.883 | 0.998 | 0.997 | 0.997 | 0.997 | 0.997 | 1.000 | 0.981 |
| ACSF | Gaussian † | 4.1 (4.0) | 1.0 (4.0) | 1.0 (1.6) | 1.0 (1.1) | 0.888 | 0.996 | 0.998 | 0.998 | 0.998 | 0.998 | 1.000 | **0.982** |
| | Laplacian † | **1.3** (1.3) | **2.0** (51.3) | **1.1** (4.7) | 1.0 (1.9) | 0.861 | 0.996 | 0.996 | 0.996 | 0.996 | 0.996 | 1.000 | 0.977 |

**Correlation Between Spectral Metrics and Performance** To test whether spectral richness translates into improved predictive accuracy, we plotted scatter plots of the four (truncated) spectral metrics against the averaged $R^2$ score across seven molecular properties (Fig. 4). We then quantified these relationships using Pearson correlation tests, reporting correlation coefficients ($\hat{r}$) and 95% confidence intervals in Table 2. Since the power-law decay parameter $\alpha$ decreases with richer spectra, we report $-\alpha$ so that a positive correlation coefficient $\hat{r}$ consistently indicates a positive relationship between spectral richness and predictive performance. The results show that the common SSL heuristic—"richer spectra yield better performance"—does not hold in general. Overall, correlations are weak, often inconclusive, and many confidence intervals span zero. For ECFP kernels, only the polynomial decay rate ($-\alpha$) displays a significant positive correlation, while the other metrics remain inconclusive. For transformer-based kernels, all correlations are negative but statistically insignificant, suggesting no reliable pattern. Global 3D kernels exhibit mixed behavior: $-\alpha$ points to a positive trend, but this is not confirmed by the other metrics. In contrast, local 3D kernels show even the opposite: all four metrics show strong *negative* correlations, with SSE and ID being statistically significant, indicating that greater spectral richness can actively hinder generalization.

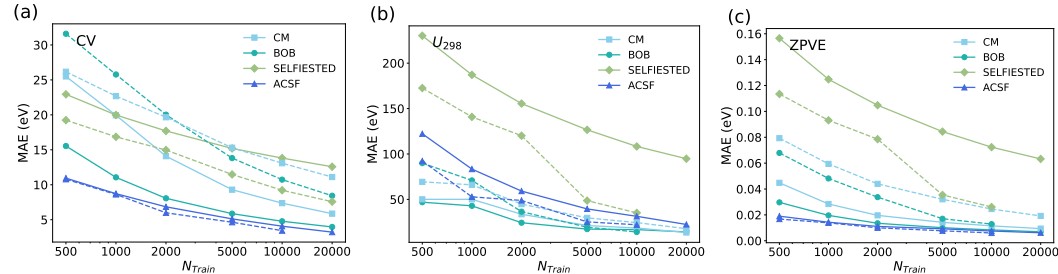

Figure 2: Test mean absolute error (MAE), computed on $10,000$ molecules, as a function of training set size for three properties: (a) $C_V$, (b) $U_{298}$, and (c) ZPVE. In all panels, results are shown for the Laplacian kernel applied to three global representations (CM, BOB, and SELFIESTED; $\sigma_\ell = 10^4$) and one local representation (ACSF; $\sigma_\ell = 100$). Solid: Laplacian, and Dashed: Gaussian kernel.

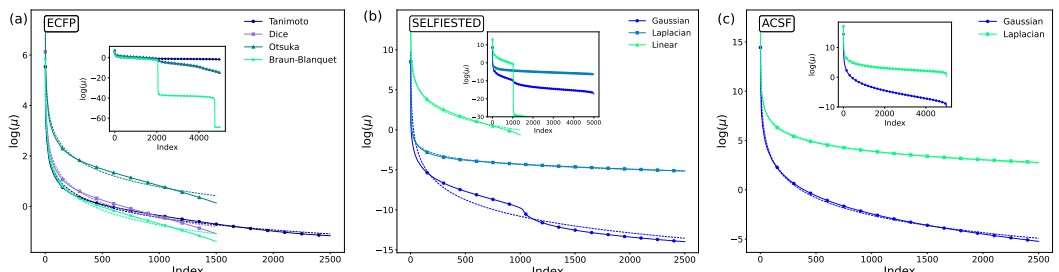

Figure 3: Kernel eigenvalue spectra with insets highlighting that nearly half of the eigenvalues are close to zero (main plots) for different molecular representations. Results are shown for (a) ECFPs, (b) SELFIESTED, and (c) local 3D descriptor-based kernels.

In summary, **spectral richness alone is not a reliable predictor of downstream performance**; its impact depends critically on the choice of molecular representation.

**Ablation on Training Size** While the results in Table 1 are reported with a fixed training size of $N_{\text{train}} = 5,000$, we also conducted an ablation study varying $N_{\text{train}}$. Fig. 2 plots the mean absolute error across different kernels and molecular properties. The results show a steady improvement in test performance as $N_{\text{train}}$ increases to 10,000 and 20,000. We expect the observation of our spectral analysis to persist for even larger training sizes.

**Truncated Kernel Ridge Regression** Moreover, we computed the TKRR (using the approximated truncated kernel in Eq. 20 at each truncation level $r$, tuning the regularization parameter independently for each case. We then record the truncation thresholds at which the performance recovers 95% and 99% of the original KRR $\mathbf{R}^2$ score (see Table 3). Note that for many kernels, retaining only the top 2% of eigenvalues recovers $> 95\%$ performance, indicating that the leading eigenvalues capture the most informative features.

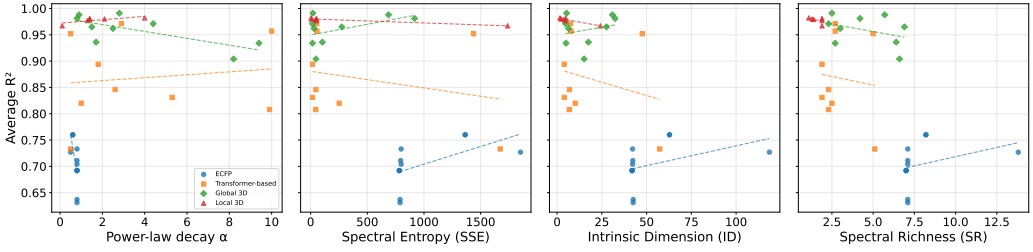

Figure 4: Correlation between spectral metrics and average $\mathbf{R}^2$ across molecular kernel categories. Dotted lines indicate the best-fit linear trend for each category.

Table 2: Pearson correlation coefficients ($\hat{r}$) with 95% confidence intervals (CI) among the spectral metrics $-\alpha$, SSE, ID, and SR, which align with the notion of spectral richness, and average $\mathbf{R}^2$ across molecular kernel categories. Correlations whose 95% CI excludes zero are shown in **bold**.

| Mol. Repr. | | $-\alpha \uparrow$ | SSE $\uparrow$ | ID$\uparrow$ | SR$\uparrow$ |
|---|---|---|---|---|---|
| ECFP | $\hat{r}$ | **0.624** | 0.582 | 0.414 | 0.334 |
| | 95% CI | [0.039,0.891] | [-0.027, 0.876] | [-0.247, 0.812] | [-0.333, 0.778] |
| Transformer-based | $\hat{r}$ | -0.129 | -0.259 | -0.255 | -0.100 |
| | 95% CI | [-0.731,0.585] | [-0.787, 0.490] | [-0.786, 0.493] | [-0.716, 0.604] |
| Global 3D | $\hat{r}$ | **0.716** | 0.474 | 0.239 | -0.373 |
| | 95% CI | [0.098, 0.935] | [-0.277, 0.866] | [-0.505, 0.780] | [-0.831, 0.387] |
| Local 3D | $\hat{r}$ | -0.755 | **-0.955** | **-0.965** | -0.559 |
| | 95% CI | [-0.971, 0.146] | [-0.995, -0.638] | [-0.996, -0.711] | [-0.943, 0.463] |

Table 3: Summary of Eigenvalue Truncation Thresholds to reach 95% and 99% of Maximum $R^2$. Global—Gaussian $\sigma_\ell = 10^2$, Laplacian $\sigma_\ell = 10^4$; Local—Gaussian and Laplacian $\sigma_\ell = 100$.

| Mol. Repr. | Kernel | Gap | | $C_V$ | | $\Delta H$ | | $U_0$ | | $U_{298}$ | | $G$ | | ZPVE | |
|---|---|---|---|---|---|---|---|---|---|---|---|---|---|---|---|
| | | 95% | 99% | 95% | 99% | 95% | 99% | 95% | 99% | 95% | 99% | 95% | 99% | 95% | 99% |
| ECFP | Tanimoto | 15.0 | 90.0 | 25.0 | 95.0 | 35.0 | 100.0 | 35.0 | 100.0 | 35.0 | 100.0 | 35.0 | 100.0 | 5.5 | 70.0 |
| | Dice | 2.9 | 9.0 | 7.2 | 30.0 | 20.0 | 35.0 | 20.0 | 35.0 | 20.0 | 35.0 | 20.0 | 35.0 | 2.9 | 15.0 |
| | Otsuka | 3.8 | 15.0 | 15.0 | 40.0 | 30.0 | 45.0 | 30.0 | 45.0 | 30.0 | 45.0 | 30.0 | 45.0 | 2.9 | 20.0 |
| | Sogenfrie | 25.0 | 100.0 | 50.0 | 100.0 | 90.0 | 100.0 | 90.0 | 100.0 | 90.0 | 100.0 | 90.0 | 100.0 | 20.0 | 100.0 |
| | Bran-Blanquet | 2.9 | 10.0 | 2.0 | 4.6 | 2.9 | 20.0 | 2.9 | 20.0 | 2.9 | 20.0 | 2.9 | 20.0 | 2.0 | 6.4 |
| | Faith | 2.9 | 10.0 | 3.8 | 40.0 | 30.0 | 45.0 | 30.0 | 45.0 | 30.0 | 45.0 | 30.0 | 45.0 | 2.9 | 15.0 |
| | Forbes | 45.0 | 7.2 | 4.6 | 30.0 | 10.0 | 50.0 | 10.0 | 50.0 | 10.0 | 50.0 | 10.0 | 50.0 | 2.9 | 8.1 |
| | Inner-Product | 2.9 | 10.0 | 2.0 | 4.6 | 2.9 | 25.0 | 2.9 | 25.0 | 2.9 | 25.0 | 2.9 | 25.0 | 2.0 | 7.2 |
| | Intersection | 2.9 | 15.0 | 3.8 | 40.0 | 30.0 | 45.0 | 30.0 | 45.0 | 30.0 | 45.0 | 30.0 | 45.0 | 2.9 | 15.0 |
| | Min-Max | 15.0 | 90.0 | 25.0 | 95.0 | 35.0 | 100.0 | 35.0 | 100.0 | 35.0 | 100.0 | 35.0 | 100.0 | 5.5 | 70.0 |
| | Rand | 2.9 | 10.0 | 3.8 | 40.0 | 30.0 | 45.0 | 30.0 | 45.0 | 30.0 | 45.0 | 30.0 | 45.0 | 2.9 | 15.0 |
| SELFIESTED | Gaussian | 15.0 | 50.0 | 2.0 | 15.0 | 2.9 | 25.0 | 2.9 | 25.0 | 2.9 | 25.0 | 2.9 | 25.0 | 2.0 | 9.0 |
| | Laplacian | 5.5 | 50.0 | 2.0 | 15.0 | 2.9 | 25.0 | 2.9 | 25.0 | 2.9 | 25.0 | 2.9 | 25.0 | 2.0 | 10.0 |
| | Linear | 5.5 | 15.0 | 2.0 | 8.1 | 3.8 | 15.0 | 3.8 | 15.0 | 3.8 | 15.0 | 3.8 | 15.0 | 2.0 | 4.6 |
| SELFormer | Gaussian | 7.2 | 25.0 | 15.0 | 40.0 | 20.0 | 45.0 | 20.0 | 45.0 | 20.0 | 45.0 | 20.0 | 45.0 | 6.4 | 25.0 |
| | Laplacian | 5.5 | 50.0 | 10.0 | 60.0 | 20.0 | 60.0 | 20.0 | 60.0 | 20.0 | 60.0 | 20.0 | 60.0 | 3.8 | 40.0 |
| | Linear | 5.5 | 15.0 | 8.1 | 15.0 | 9.0 | 95.0 | 9.0 | 95.0 | 9.0 | 95.0 | 9.0 | 95.0 | 5.5 | 15.0 |
| MLT-BERT | Gaussian | 10.0 | 40.0 | 7.2 | 35.0 | 10.0 | 30.0 | 10.0 | 30.0 | 10.0 | 30.0 | 10.0 | 30.0 | 6.4 | 85.0 |
| | Laplacian | 15.0 | 45.0 | 15.0 | 45.0 | 20.0 | 65.0 | 20.0 | 65.0 | 20.0 | 65.0 | 20.0 | 65.0 | 7.2 | 40.0 |
| | Linear | 2.9 | 4.6 | 2.9 | 4.6 | 2.9 | 4.6 | 2.9 | 4.6 | 2.9 | 4.6 | 2.9 | 4.6 | 2.0 | 40.0 |
| CM | Gaussian | 20.0 | 55.0 | 2.9 | 30.0 | 2.0 | 2.9 | 2.0 | 2.9 | 2.0 | 2.9 | 2.0 | 2.9 | 2.0 | 2.9 |
| | Laplacian | 25.0 | 70.0 | 7.2 | 25.0 | 2.0 | 2.9 | 2.0 | 2.9 | 2.0 | 2.9 | 2.0 | 2.9 | 2.0 | 2.9 |
| | Linear | 2.0 | 5.5 | 2.0 | 5.5 | < 0.1 | 4.6 | < 0.1 | 4.6 | < 0.1 | 4.6 | < 0.1 | 4.6 | 2.0 | 5.5 |
| BOB | Gaussian | 20.0 | 40.0 | 8.1 | 30.0 | 2.0 | 2.0 | 2.0 | 2.0 | 2.0 | 2.0 | 2.0 | 2.0 | 2.0 | 3.8 |
| | Laplacian | 10.0 | 50.0 | 2.9 | 9.0 | 2.0 | 2.0 | 2.0 | 2.0 | 2.0 | 2.0 | 2.0 | 2.0 | 2.0 | 2.0 |
| | Linear | 2.0 | 3.8 | 2.9 | 5.5 | 2.0 | 2.0 | 2.0 | 2.0 | 2.0 | 2.0 | 2.0 | 2.0 | 2.0 | 2.0 |
| SLATM | Gaussian | 5.5 | 20.0 | 2.0 | 2.0 | 2.0 | 2.0 | 2.0 | 2.0 | 2.0 | 2.0 | 2.0 | 2.0 | 0.2 | 2.0 |
| | Laplacian | 7.2 | 100.0 | 2.0 | 5.5 | 2.0 | 4.6 | 2.0 | 4.6 | 2.0 | 4.6 | 2.0 | 4.6 | 2.0 | 2.0 |
| | Linear | 10.0 | 20.0 | 2.0 | 2.0 | 2.0 | 2.0 | 2.0 | 2.0 | 2.0 | 2.0 | 2.0 | 2.0 | 0.1 | 2.0 |
| SOAP | Gaussian | 75.0 | 35.0 | 85.0 | 75.0 | 0.1 | 0.1 | 0.1 | 0.1 | 0.1 | 0.1 | 0.1 | 0.1 | < 0.1 | < 0.1 |
| | Laplacian | 15.0 | 35.0 | 2.0 | 2.9 | 2.0 | 2.0 | 2.0 | 2.0 | 2.0 | 2.0 | 2.0 | 2.0 | < 0.1 | 0.1 |
| FCHL19 | Gaussian | 4.6 | 15.0 | 2.0 | 2.0 | 2.0 | 2.0 | 2.0 | 2.0 | 2.0 | 2.0 | 2.0 | 2.0 | < 0.1 | 0.1 |
| | Laplacian | 15.0 | 50.0 | 2.0 | 2.0 | 2.0 | 2.9 | 2.0 | 2.9 | 2.0 | 2.9 | 2.0 | 2.9 | < 0.1 | 0.1 |
| ACSF | Gaussian | 10.0 | 30.0 | 2.0 | 2.9 | 2.0 | 2.0 | 2.0 | 2.0 | 2.0 | 2.0 | 2.0 | 2.0 | < 0.1 | 0.1 |
| | Laplacian | 20.0 | 70.0 | 2.0 | 4.6 | 2.0 | 6.4 | 2.0 | 6.4 | 2.0 | 6.4 | 2.0 | 6.4 | 0.1 | 0.2 |

## 4 DISCUSSION

In this section, we discuss the main implications of our empirical findings for (i) kernel theory and self-supervised learning, (ii) practical molecular-chemistry practice, and (iii) limitations and avenues for future work.

## 4.1 Insights for Kernel Theory and Self-Supervised Learning

**Fingerprint Kernel**   It is surprising to see that ECFP-based kernels are the only category showing a slight positive correlation between spectral richness and predictive performance. This aligns with the long-observed empirical fact that the Tanimoto kernel, the preferred kernel in cheminformatics, often outperforms other ECFP-based kernels like Dice and Otsuka in downstream tasks. Our spectral analysis provides a principled explanation: the key difference lies in the richness of the spectral tail, with Tanimoto retaining more information in the lower-ranked eigenvectors (see Fig. 5). In this narrow setting, the common SSL heuristic—"richer spectra yield better performance"—appears to hold. However, this intuition breaks down when considering the Sogenfrei kernel, which possesses the richest spectrum among ECFP kernels but delivers only average performance. This might suggest that ECFP-based kernels, being hand-designed, may be fundamentally different from SSL-derived features: they already encode domain knowledge in the representation itself, so most relevant information is concentrated in the top eigenvectors, making spectral richness less decisive.

**Pretrained Transformers**   Moreover, for transformer-based kernels, where representations are generated from models pretrained on large chemical corpora and then evaluated on unseen QM9 tasks in a setup analogous to SSL, the heuristic fails even more clearly: correlations are consistently negative, albeit weak. For 3D global kernels, the evidence remains inconclusive. By contrast, 3D local kernels show a strong and systematic negative correlation across all spectral metrics, with Gaussian kernels often outperforming Laplacian kernels despite having a faster spectral decay. This inversion of the heuristic highlights that spectral richness can, in fact, be detrimental, depending on the kernel and representation. The underlying reason remains unclear, but it opens an intriguing direction for both kernel theory and materials science.

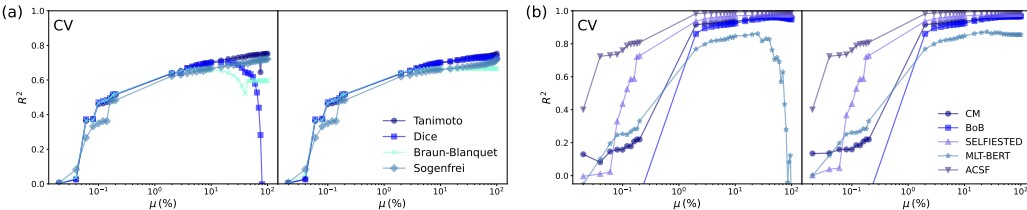

Figure 5: $R^2$ score for the heat capacity ($C_V$) property as a function of truncation level ($\mu(\%)$ for (a) selected ECFP-based kernels and (b) four various global (CM, BOB, SELFIESTED, MLT-BERT) kernel and a single local (ACSF) kernel, all with a Gaussian kernel with $\sigma_\ell = 100$). Left sub-panel: without regularization; right sub-panel: with regularization.

**Regularization versus Truncation**   Regularization has long been a standard technique in machine learning to mitigate overfitting to label noise. In kernel methods, it works by penalizing the use of high-frequency eigenfunctions in fitting the data. Interestingly, truncation achieves a similar effect by explicitly discarding the tail of the spectrum, thereby removing high-frequency components from the hypothesis space. As shown in Fig. 5 and Figs. 10-12 in Section B.2, the performance of the best ridgeless truncated KRR (left panel) is comparable to that of the fully regularized KRR (right panel). This observation provides a possible explanation for why richer spectra may sometimes harm generalization: additional eigenfunctions in the tail can facilitate overfitting rather than improve predictive accuracy, and any regularization to avoid overfitting would harm the accuracy. Notably, this phenomenon is not unique to ECFP-based kernels, but is also observed across other kernel categories (see Section B.2 for additional plots). To the best of our knowledge, there is currently no theoretical work establishing a formal connection between truncation and regularization, making this a promising direction for future research in machine learning theory.

## 4.2 Insights for Molecular Chemistry

**First Comprehensive Results**   Pretrained molecular embedding models have recently attracted significant interest in chemistry, particularly for small molecules, as they are increasingly adopted for tasks such as drug design. Related work has applied pretrained embeddings in a kernel framework for proteins; however, these efforts were limited to kernel construction without further spectral analysis, such as ours. In contrast, this work is the first to explore a kernel-based framework built

upon pretrained molecular embedding models for chemistry while also analyzing their spectral characteristics.

**Transformer-based Representations**  Previous work has mainly applied linear or MLP-based regression to transformer-derived molecular representations Praski et al. (2025), motivated by the high dimensionality of embeddings, where kernel matrices often resemble their linearization—a weighted sum of the covariance matrix, identity, and a rank-one term El Karoui (2010). In contrast, we show that kernel ridge regression with a Gaussian kernel outperforms the linear baseline, indicating that higher-order terms capture additional information beyond linear covariance. Notably, SELFIESTED with a Gaussian kernel achieved the best performance. This suggests an alternative way to evaluate SSL models—via spectral metrics derived from their kernel matrices—which we leave for future work. We also evaluated ChemBERTa but found consistently weaker performance than ECFP-based kernels. For instance, on QM9 with Gaussian/Laplacian kernels, $R^2$ scores were 0.201/0.193 for GAP, 0.102/0.094 for $U_0$, and 0.247/0.249 for $C_V$. Given its poor results, we omit ChemBERTa from Table 1.

**3D Descriptors**  The comparison between global and local 3D kernels, whose representations are built on Cartesian coordinates, has sparked the latter development in molecular kernels Thant et al. (2025). However, we found that global 3D kernels are more susceptible to drastic effects in their accuracy when hyperparameter search is found to be suboptimal, contrary to local 3D kernels. For global 3D representations, SLATM consistently outperformed other descriptors regardless of kernel choice; notably, even the linear kernel with SLATM surpassed the CM–based kernel in accuracy. Finally, local 3D representations were found to be the most consistent across kernels and also delivered the overall highest scores, except in the case of SLATM combined with Gaussian or Laplacian kernels, which remained competitive.

### 4.3 LIMITATIONS AND FUTURE WORK

Despite our systematic experiments and analyses, several limitations remain.

**Data**  While our study is limited to QM9, this dataset remains one of the most widely adopted benchmarks for molecular property prediction Ramakrishnan et al. (2014); Gilmer et al. (2017); Schütt et al. (2017); Wu et al. (2018). Its ∼134k molecules cover a chemically diverse space of small organics and provide DFT-computed properties across multiple thermodynamic and electronic targets, which makes it an ideal controlled testbed for comparative studies. QM9 continues to serve as a standard proxy in both kernel-based Faber et al. (2018); Christensen et al. (2020) and NN–based approaches Schütt et al. (2017); Thomas et al. (2018), precisely because it allows systematic exploration of representations and models without confounding experimental noise or inconsistencies across datasets.

**Representations and Kernels**  We did not include recent graph-based encoders such as GROVER Rong et al. (2020) or hybrid approaches like Mol2Vec Jaeger et al. (2018), which may reveal distinct spectral behaviors. Similarly, quantum-inspired kernels derived from molecular graph circuits Schuld et al. (2020); Torabian & Krems (2025) represent another promising direction for applying our framework to evaluate the structure and capacity of emerging methods in chemistry and materials science.

### 5 CONCLUSION

We presented the first systematic spectral analysis of molecular kernels for property prediction on QM9, spanning kernels with ECFP, pretrained transformer-based features, and global or local 3D descriptors as inputs. Our results show that spectral richness is not a universal predictor of performance: by the Pearson test, it correlates negatively with transformer-based and local 3D kernels and remains inconsistent for global 3D and ECFP representations. The truncated kernels revealed that in many kernels, retaining only the top 2% of the eigenvalues is often sufficient to recover  95% of the original precision. These findings call into question the common heuristic that "richer spectra yield better generalization." More broadly, our study offers practical guidance for pairing molecular representations with kernels and opens a new avenue for bridging spectral analysis between self-supervised learning and kernel methods.

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

# Appendix

The appendix is organized as follows. Section A introduces the molecular representations and kernel functions considered in this work, including fingerprint-based kernels, pretrained text-embedding models, and Cartesian coordinate–based representations. As noted in the main text, we use the term **3D** kernels to refer to kernels derived from Cartesian coordinates, whether global or local. Section B presents supplementary experimental results. Section C provides detailed definitions of the four spectral metrics. Finally, Section D contains proofs omitted from the main text.

## A    MOLECULAR KERNELS

Here, we briefly summarize molecular kernels that are based on molecular representations, which can broadly be divided into two categories:

**Definition 1** (Global Molecular Representation). *Let $\mathcal{M}$ denote a molecule and $\phi : \mathcal{M} \to \mathbb{R}^d$ be a function that maps a molecule to a $d$-dimensional vector of descriptors that summarize the entire structure (e.g., fingerprints, Coulomb matrix eigenvalues, or learned embeddings by encoding models).*

**Definition 2** (Local Molecular Representation). *Let $\mathcal{M}$ denote a molecule composed of Na atoms, where each atom is represented by $z_\ell$ containing Cartesian coordinates and nuclear information such as atomic number. A local representation is given by a function $\phi_\ell : z_\ell \mapsto \mathbb{R}^d$ that encodes atomic environments based on the arrangement of neighboring atoms. Examples include the Smooth Overlap of Atomic Positions and many-body distribution functions.*

Due to the existence of $\phi$ and $\phi_\ell$, there are two main families of molecular kernels: global and local molecular kernels.

**Definition 3** (Global Molecular Kernel). *A* global molecular kernel *is a positive-definite function $k_{\text{global}} : \mathcal{M}_i \times \mathcal{M}_j \to \mathbb{R}$ defined as*

$$k_{global}(\mathcal{M}_i, \mathcal{M}_j) = \kappa\big(\phi(\mathcal{M}_i), \phi(\mathcal{M}_j)\big), \tag{3}$$

*where $\kappa : \mathbb{R}^d \times \mathbb{R}^d \to \mathbb{R}$ is a positive-definite kernel function comparing global descriptor vectors computed with $\phi$.*

A prominent example of a global kernel is obtained when $\phi$ is computed via extended connectivity fingerprints (ECFPs) Rogers & Hahn (2010). ECFPs are fixed-length hashed descriptors generated by iteratively encoding atom-centered circular neighborhoods (the Morgan algorithm) up to a pre-defined radius $r$. The resulting binary vector, $z_i^\top = \phi_{\text{ECFP}}(\mathcal{M}_i)^\top = [1, 0, 1, \cdots, 1]^\top$, captures the 2D molecular topology (and, optionally, chirality) in a global form. When using $\phi_{\text{ECFP}}(\mathcal{M})$ as the descriptor, similarity can be quantified through fingerprint-specific kernels such as

$$k_{\text{Tanimoto}}(\mathcal{M}_i, \mathcal{M}_j) = \sigma_f^2 \cdot \frac{\langle \boldsymbol{x}_i, \boldsymbol{x}_j \rangle}{|\boldsymbol{x}_i|_2^2 + |\boldsymbol{x}_j|_2^2 - \langle \boldsymbol{x}_i, \boldsymbol{x}_j \rangle}, \quad k_{\text{Dice}}(\mathcal{M}_i, \mathcal{M}_j) = \sigma_f^2 \cdot \frac{2\langle \boldsymbol{x}_i, \boldsymbol{x}_j \rangle}{|\boldsymbol{x}_i|_1 + |\boldsymbol{x}_j|_1}, \tag{4}$$

where $\sigma_f$ is a kernel hyperparameter, $\boldsymbol{x} = \phi_{\text{ECFP}}(\mathcal{M})$, $\langle \boldsymbol{x}_i, \boldsymbol{x}_j \rangle = \boldsymbol{x}_i^\top \boldsymbol{x}_j$, and $|\boldsymbol{x}_j|_p$ is the $p$-norm of the fingerprint vector. We present other ECFP-based kernels in Section A.1.1.

Another widely used class of global descriptors arises from *data-driven molecular embeddings*, where $\phi$ is learned from large corpora of molecular strings such as SMILES or SELFIES. Examples include models such as SELFIESTED, SELFormer, and MLT-BERT, which leverage transformer-based language models to capture chemical semantics. Unlike discrete fingerprints, these embeddings yield continuous-valued feature vectors, enabling the use of standard isotropic kernels such as Gaussian, Laplacian, or linear. Additional details of transformer-based global representations are provided in Section A.1.2.

Beyond data-driven embeddings, global representations can also incorporate explicit geometrical information. A classical example is the Coulomb Matrix (CM) Rupp et al. (2012), which encodes pairwise Coulombic interactions between atoms. Other notable global descriptors include bag of

bonds (BoB) Hansen et al. (2015) and the spectrum of London and Axilrod–Teller–Muto (SLATM) Huang & von Lilienfeld (2020). BoB, inspired by the bag-of-words algorithm in natural language processing, extends the CM by grouping pairwise interactions into "bags" according to bond type. SLATM, in contrast, is based on many-body expansions: it represents molecular structures by approximating atomic charge densities with Gaussian functions scaled by interatomic potentials. Additional details of local kernels are provided in Section A.1.

Although global descriptors capture holistic molecular information, they may struggle to generalize across molecules with different sizes or conformations. Much of the recent work on molecular kernel development has therefore focused on incorporating geometric information at the atomic level. *Local kernels* address this by encoding atomic environments within a cutoff radius, making them naturally suited to enforce invariances (e.g., translation, rotation, and permutation) and improving transferability across chemical space. Prominent examples include the Smooth Overlap of Atomic Positions (SOAP) Bartók et al. (2013), Faber–Christensen–Huang–Lilienfeld (FCHL) Faber et al. (2018); Christensen et al. (2020), and many-body distribution functions (MBDF) Khan et al. (2023); Khan & von Lilienfeld (2024).

**Definition 4** (Local Molecular Kernel). *A* local molecular kernel *is a positive-definite function of two molecules, defined as*

$$k_{local}(\mathcal{M}_i, \mathcal{M}_j) = \sum_{\ell_i=1}^{Na_i} \sum_{\ell_j=1}^{Na_j} g(Z_{\ell_i}, Z_{\ell_j})\, \kappa\big(\phi_\ell(\boldsymbol{z}_{\ell_i}), \phi_\ell(\boldsymbol{z}_{\ell_j})\big),$$

(5)

*where $\boldsymbol{z}_{\ell_i}$ denotes the position and chemical identity of the $\ell_i$-th atom in $\mathcal{M}_i$, $\phi_\ell$ maps its local chemical environment to a descriptor (e.g., SOAP, FCHL19, ACSF), and $\kappa$ is a positive-definite kernel function (such as Gaussian or Laplacian) that measures similarity between atomic environments. The function $g(Z_{\ell_i}, Z_{\ell_j})$ compares atomic species, typically defined as a Kronecker delta on the atomic numbers, i.e. $g(Z_{\ell_i}, Z_{\ell_j}) = \delta(Z_{\ell_i} = Z_{\ell_j})$.*

## A.1 GLOBAL MOLECULAR KERNELS/REPRESENTATIONS

### A.1.1 EXTENDED-CONNECTIVITY FINGER PRINTS

One of the most common global molecular representations is the extended-connectivity fingerprints (ECFPs) Rogers & Hahn (2010). Here is a list of some of the global molecular kernels based on the ECFP representation,

$$k_{\text{Braun-Blanquet}} = \frac{\langle \boldsymbol{x}_1, \boldsymbol{x}_2 \rangle}{\max(|\boldsymbol{x}_1|, |\boldsymbol{x}_2|)}, \qquad k_{\text{Dice}} = \frac{2\langle \boldsymbol{x}_1, \boldsymbol{x}_2 \rangle}{|\boldsymbol{x}_1| + |\boldsymbol{x}_2|}$$

(6)

$$k_{\text{Faith}} = \frac{2\langle \boldsymbol{x}_1, \boldsymbol{x}_2 \rangle + d_0}{2d}, \qquad k_{\text{Forbes}} = \frac{d\langle \boldsymbol{x}_1, \boldsymbol{x}_2 \rangle}{|\boldsymbol{x}_1| + |\boldsymbol{x}_2|}$$

(7)

$$k_{\text{Inner-Product}} = \langle \boldsymbol{x}_1, \boldsymbol{x}_2 \rangle = \boldsymbol{x}_1^\top \boldsymbol{x}_2, \qquad k_{\text{Intersection}} = \langle \boldsymbol{x}_1, \boldsymbol{x}_2 \rangle + \langle \boldsymbol{x}_1', \boldsymbol{x}_2' \rangle$$

(8)

$$k_{\text{MinMax}} = \frac{|\boldsymbol{x}_1| + |\boldsymbol{x}_2| - |\boldsymbol{x}_1 - \boldsymbol{x}_2|}{|\boldsymbol{x}_1| + |\boldsymbol{x}_2| + |\boldsymbol{x}_1 - \boldsymbol{x}_2|}, \qquad k_{\text{Otsuka}} = \frac{\langle \boldsymbol{x}_1, \boldsymbol{x}_2 \rangle}{\sqrt{|\boldsymbol{x}_1| + |\boldsymbol{x}_2|}}$$

(9)

$$k_{\text{Rogers-Tanimoto}} = \langle \boldsymbol{x}_1, \boldsymbol{x}_2 \rangle + \frac{d_0}{2|\boldsymbol{x}_1|} + 2|\boldsymbol{x}_2| - 3\langle \boldsymbol{x}_1, \boldsymbol{x}_2 \rangle + d_0, \qquad k_{\text{Rand}} = \frac{\langle \boldsymbol{x}_1, \boldsymbol{x}_2 \rangle + d}{n}$$

(10)

$$k_{\text{Russel-Roa}} = \frac{\langle \boldsymbol{x}_1, \boldsymbol{x}_2 \rangle}{n}, \qquad k_{\text{Sogenfei}} = \frac{\langle \boldsymbol{x}_1, \boldsymbol{x}_2 \rangle^2}{|\boldsymbol{x}_1| + |\boldsymbol{x}_2|}$$

(11)

$$k_{\text{Soakl-Sneath}} = \frac{\langle \boldsymbol{x}_1, \boldsymbol{x}_2 \rangle}{2|\boldsymbol{x}_1|} + 2|\boldsymbol{x}_2| - 3\langle \boldsymbol{x}_1, \boldsymbol{x}_2 \rangle, \qquad k_{\text{Tanimoto}} = \frac{\langle \boldsymbol{x}_1, \boldsymbol{x}_2 \rangle}{\|\boldsymbol{x}_1\|^2 + \|\boldsymbol{x}_2\|^2 - \langle \boldsymbol{x}_1, \boldsymbol{x}_2 \rangle}$$

(12)

where:

- $x_i$ is the global representation of the molecule using the ECFPs; $x_i = \phi_{\text{ECFP}}(\mathcal{M}_i)$, for example, $x_i^\top = [1, 0, 1, \cdots, 1]^\top$.

- $\langle x_i, x_j \rangle$ denotes the inner product.

- $x_i'$ is the bit-flipped vector of $x_i$.

- $|x_i|$ represents the $L_1$ norms of $x_i$.

- $d_0$ is the number of common zeros, and $d$ is the dimension of the input vectors,

### A.1.2 Pretrained Molecular Embedding Models

Pretrained molecular embedding models Praski et al. (2025) have become a standard approach for molecular property prediction. These models are trained on large molecular corpora to produce embedding vectors $z \in \mathbb{R}^d$, which can then be used for downstream regression tasks. We briefly describe the three pretrained transformer-based models used in our analysis:

- `SELFIESTED`: a BART-based encoder–decoder model for SELFIES, with 358M parameters, 12 layers, and 16 attention heads Priyadarsini et al. (2025); `ibm/materials.selfies-ted`.

- `SELFormer`: a RoBERTa-style encoder-only model for SELFIES, with 86M parameters, 12 layers, and 4 attention heads Yüksel et al. (2023). `HUBioDataLab/SELFormer`

- `MLT-BERT`: a BERT-style transformer model for sequence modeling, with 16M parameters, 8 layers, and 8 attention heads Zhang et al. (2022). `jonghyunlee/ChemBERT_ChEMBL_pretrained`

- `ChemBERTa`: a RoBERTa-style encoder-only model for SMILES, with 10M parameters, 6 layers, and 12 attention heads (72 attention mechanisms in total) Chithrananda et al. (2020). `Phando/chemberta-v2-finetuned-uspto-50k-classification`

For these global text embedding models, denoted $\phi_{\text{LLM}}$, we evaluated three kernel functions: linear, isotropic Gaussian, and isotropic Laplacian.

### A.1.3 Global Cartesian Coordinates Molecular Representations

**CM Rupp et al. (2012)**: CM is a global descriptor that encodes pairwise electrostatic interactions between atoms:

$$C_{ij} = \begin{cases} 0.5 Z_i^{2.4} & \text{if } i = j \\ \frac{Z_i Z_j}{R_{ij}} & \text{if } i \neq j, \end{cases} \tag{13}$$

where where $Z_i$ is the atomic number of atom $i$ and $R_{ij}$ is the interatomic distance; $R_{ij} = \|\mathbf{R}_i - \mathbf{R}_j\|$. Despite its simplicity, CM is not invariant to atom indexing, which limits its generalization. To ensure invariance to atom indexing, each molecule is represented via the eigenvalue spectrum of its CM, sorted by descending absolute value. This diagonalized form is invariant to permutations, translations, and rotations, and yields a continuous molecular distance metric even for molecules with different numbers of atoms (using zero-padding).

**Bag of Bonds (BoB) Hansen et al. (2015)**: The BoB descriptor is inspired by the bag-of-words model from natural language processing, yielding rotational, translational, and permutation-invariant molecular representations. BoB extends the Coulomb Matrix by grouping pairwise atomic interactions into "bags" based on bond types, with each entry computed as $Z_i Z_j / |R_i - R_j|$. The entries in each bag are sorted by magnitude and zero-padded for consistent vector length. While effective for machine learning tasks, BoB cannot distinguish between homometric molecules.

**Spectrum of London and Axilrod–Teller–Muto (SLATM) Huang & von Lilienfeld (2020)**: LATM builds on many-body expansions to describe molecular structures. It models atomic environments by approximating charge densities with Gaussian functions scaled by interatomic potentials. The representation captures one-body (atomic type), two-body (pairwise distances via a London-like potential), and three-body (angles via the Axilrod–Teller–Muto potential) interactions. Each term is binned into histograms to produce fixed-length atomic vectors, ensuring invariance to translation, rotation, and permutation. SLATM supports both local (atomic-level) representations and global ones formed by summing over atomic vectors, making it effective for a wide range of molecular machine learning tasks.

### A.2 LOCAL MOLECULAR KERNELS

We considered three widely used local molecular kernels:

- **Smooth Overlap of Atomic Positions (SOAP)** Bartók et al. (2013).

- **Faber–Christensen–Huang–Lilienfeld 2019 (FCHL19)** Christensen et al. (2020).

- **Atom-Centered Symmetry Functions (ACSF)** Behler (2011).

As in prior works Faber et al. (2018); Christensen et al. (2020); Khan et al. (2023); Khan & von Lilienfeld (2024), local kernels are constructed using an element-matching function,

$$g(Z_i, Z_j) = \delta(Z_i = Z_j),$$

so that, in **Definition** 4, only atoms of the same chemical species in molecules $\mathcal{M}_i$ and $\mathcal{M}_j$ contribute to the kernel evaluation. In our experiments, all Gaussian and Laplacian kernels built on local representations follow this convention. The resulting local kernel takes the form

$$k_{\mathrm{local}}(\mathcal{M}_i, \mathcal{M}_j) = \sum_{\ell_i=1}^{\mathrm{Na}_i} \sum_{\ell_j=1}^{\mathrm{Na}_j} \delta(Z_{\ell_i} = Z_{\ell_j})\, \kappa\big(\phi_\ell(\boldsymbol{z}_{\ell_i}), \phi_\ell(\boldsymbol{z}_{\ell_j})\big), \tag{14}$$

where $\phi_\ell(\boldsymbol{z}_\ell)$ denotes the local atomic descriptor (e.g., SOAP, FCHL19, or ACSF), and $\kappa$ is either an isotropic Gaussian or Laplacian kernel.

## B  ADDITIONAL RESULTS

### B.1 KERNEL EIGENVALUE SPECTRA

Figs. 6–9 are the eigenvalue spectra of various global and local kernels.

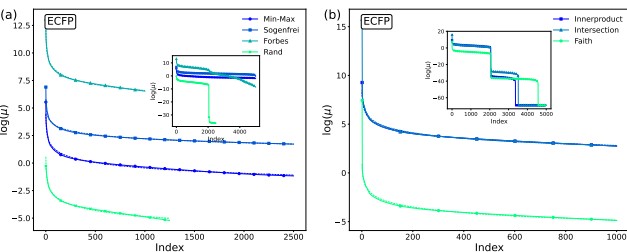

Figure 6: Kernel eigenvalue spectra with insets highlighting that nearly half of the eigenvalues are close to zero (main plots) for different ECFP-based kernels.

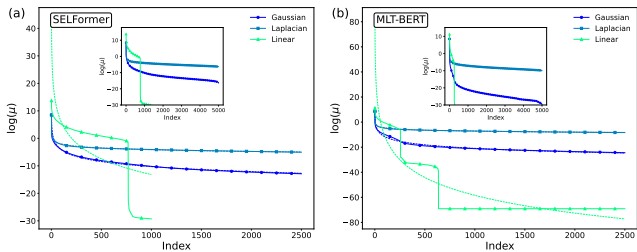

Figure 7: Kernel eigenvalue spectra with insets highlighting that nearly half of the eigenvalues are close to zero (main plots) for (a) SELFormer-based and (b) MLT-BERT-based kernels.

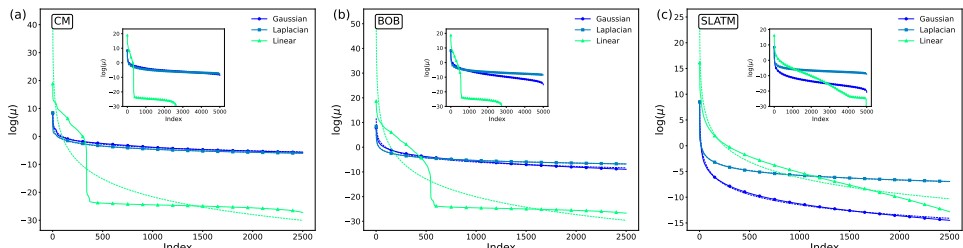

Figure 8: Kernel eigenvalue spectra with insets highlighting that nearly half of the eigenvalues are close to zero (main plots) for (a) CM, (b) BOB, (c) SLATM global representations. For all, we considered the Gaussian, Laplacian, and linear kernels.

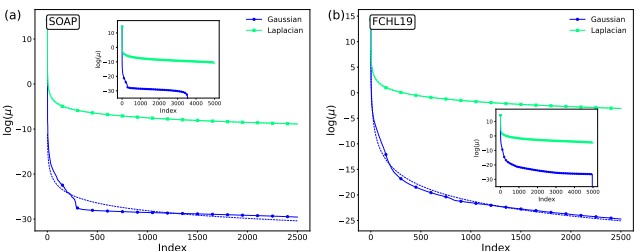

Figure 9: Kernel eigenvalue spectra with insets highlighting that nearly half of the eigenvalues are close to zero (main plots) for (a) SOAP, and (b) FCHL19 3D local representations. For both, we considered the Gaussian and Laplacian kernels.

### B.2 Truncation versus no Truncation

Figs. 10–12 represent the $\mathbf{R}^2$ score, for a test set of $10,000$ molecules, for various properties when different truncation levels are considered. At each truncation level, all hyperparameters were optimized. Fig. 10 presents the results for four ECFP-based kernels , Fig. 11 for four global representations (CM, BOB, SELFIESTED, and MLT-BERT), all using the Gaussian kernel, and Fig. 12 for three local representations (CSOAP, FCHL19, ACSF), all using the Gaussian kernel.

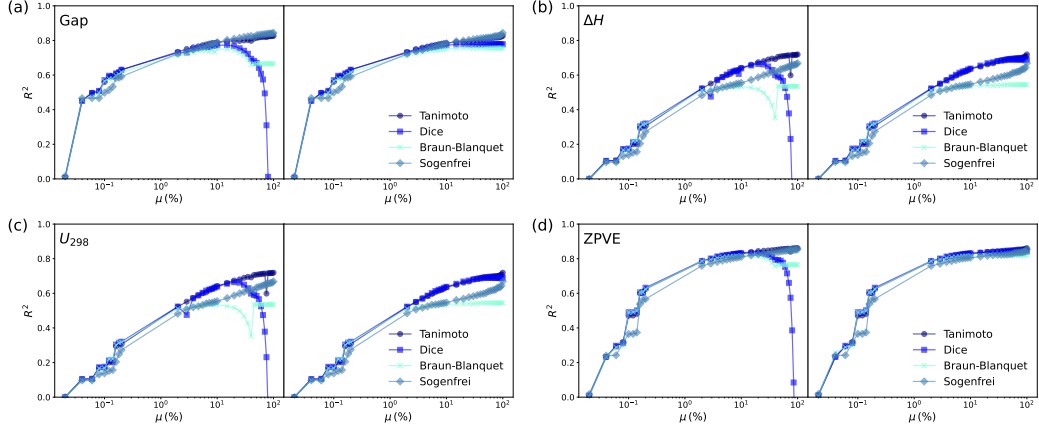

Figure 10: $\mathbf{R}^2$ score for various properties as a function of truncation level for selected ECFP-based kernels. Left and right subpanels only consider results without and with regularization, respectively. (a) Gap, (b) $\Delta H$, (c) $U_{298}$, and (d) ZPVE.

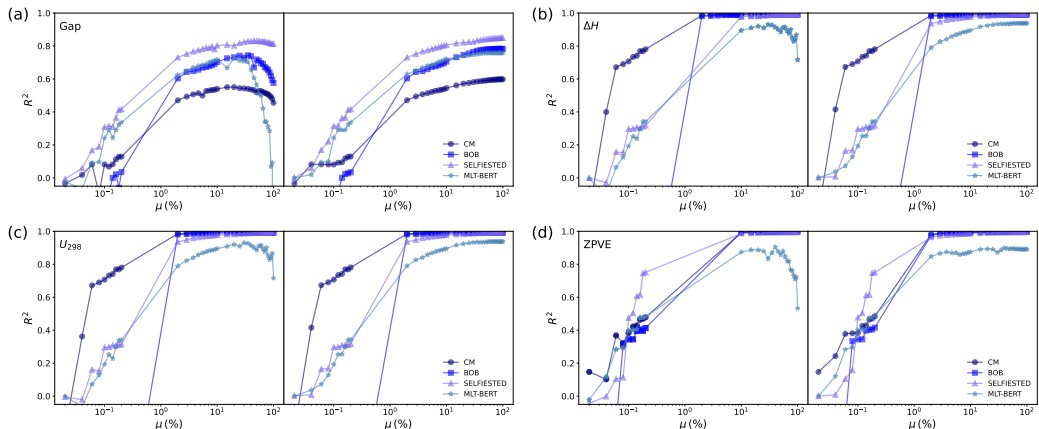

Figure 11: $\mathbf{R}^2$ score for various properties as a function of truncation level for four global representations and the Gaussian kernel with $\sigma_\ell = 100$. Left and right subpanels only consider results without and with regularization, respectively. (a) Gap, (b) $\Delta H$, (c) $U_{298}$, and (d) ZPVE.

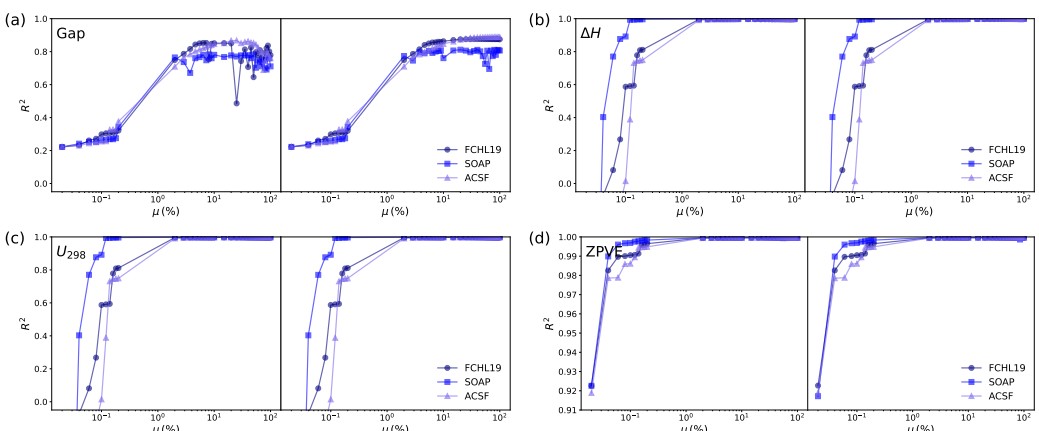

Figure 12: $\mathbf{R}^2$ score for various properties as a function of truncation level for the three local representations and the Gaussian kernel with $\sigma_\ell = 100$. Left and right subpanels only consider results without and with regularization, respectively. (a) Gap, (b) $\Delta H$, (c) $U_{298}$, and (d) ZPVE.

## C    SPECTRAL METRICS

**Definition 5** (Power Law Decay or polynomial decay rate). *Let $\{\mu_1, \mu_2, \ldots, \mu_p\}$, $p \in \mathbb{N} \cup \{\infty\}$ denote a non-increasing spectrum of positive values. We say the spectrum exhibits a* power law decay *if there exists an exponent $\alpha > 0$ such that*

$$\mu_j \propto j^{-\alpha}, \quad j = 1, 2, \ldots, p. \tag{15}$$

*The decay rate $\alpha$ can be estimated empirically by performing a linear regression on the log-log plot of the spectrum, i.e., $\log \mu_j \approx -\alpha \log j$ Agrawal et al. (2022); Mallinar et al. (2022).*

**Definition 6** (p-Stable rank). *Suppose the integers $m \geq n$ and the matrix $\boldsymbol{A} \in \mathbb{R}^{m \times n}$ has singular values $s_1(\boldsymbol{A}) \geq s_2(\boldsymbol{A}) \geq \ldots \geq s_n(\boldsymbol{A})$. For $1 \leq p \leq \infty$, the $p$-Schatten norm is defined to be*

$$\|\boldsymbol{A}\|_p \overset{\text{def.}}{=} \sqrt[p]{s_1(\boldsymbol{A})^p + \ldots + s_n(\boldsymbol{A})^p}. \tag{16}$$

*And the $p$-stable rank of the matrix $\boldsymbol{A}$ is defined to be*

$$r_p(\boldsymbol{A}) \overset{\text{def.}}{=} \frac{\|\boldsymbol{A}\|_p^p}{\|\boldsymbol{A}\|_{op}^p}. \tag{17}$$

**Definition 7** (Intrinsic dimension (ID) and stable rank (SR)). *Note that the notation of $p$-stable rank unifies the two metrics intrinsic dimension and stable rank, which are often used in ill-conditioned matrices. In particular, we have*

$$r_1(\boldsymbol{A}) = \frac{\|\boldsymbol{A}\|_1}{\|\boldsymbol{A}\|_{op}} = \frac{s_1(\boldsymbol{A}) + \ldots + s_n(\boldsymbol{A})}{s_1(\boldsymbol{A})} = \textit{intrinsic dimension of } \boldsymbol{A};$$

$$r_2(\boldsymbol{A}) = \frac{\|\boldsymbol{A}\|_2^2}{\|\boldsymbol{A}\|_{op}^2} = \frac{\|\boldsymbol{A}\|_F^2}{\|\boldsymbol{A}\|_{op}^2} = \textit{stable rank of } \boldsymbol{A}.$$

In particular, the true rank of $\boldsymbol{A}$ is always an upper bound of $r_p(\boldsymbol{A})$ for any $p$. In particular,

**Proposition 8** (Remark 5.4 in Ipsen & Saibaba (2024)). *Suppose the integers $m \geq n$ and $p \geq q$. Then for any matrix $\boldsymbol{A} \in \mathbb{R}^{m \times n}$, we have*

$$1 \leq r_p(\boldsymbol{A}) \leq r_q(\boldsymbol{A}) \leq rank(\boldsymbol{A}) \leq n. \tag{18}$$

We notice that there is another measure of rank used in ML literature:

**Definition 9** (Spectral Shannon Entropy (SSE), Definition 2.1 in Huh et al. (2023)). *Suppose the integers $m \geq n$ and the matrix $\boldsymbol{A} \in \mathbb{R}^{m \times n}$ has singular values $s_1(\boldsymbol{A}) \geq s_2(\boldsymbol{A}) \geq \ldots \geq s_n(\boldsymbol{A})$. Let $\bar{s}_i(\boldsymbol{A}) \overset{\text{def.}}{=} \frac{s_i(\boldsymbol{A})}{s_1(\boldsymbol{A}) + \ldots + s_n(\boldsymbol{A})}$ be the normalized singular values such that $\bar{s}_1(\boldsymbol{A}) + \ldots + \bar{s}_n(\boldsymbol{A}) = 1$. The spectral entropy, or the effective dimension, of $\boldsymbol{A}$ is defined to be:*

$$\rho(\boldsymbol{A}) \overset{\text{def.}}{=} \exp(-\sum_{i=1}^{n} \bar{s}_i(\boldsymbol{A}) \log(\bar{s}_i(\boldsymbol{A}))). \tag{19}$$

## D    PROOF

In this section, we present the proof which are omitted in the main text.

**Theorem 10.** *With notation above, let $\hat{f}$ be the KRR predictor in Eq. (1) and $\hat{f}^{(r)}$ the TKRR predictor with truncation level $r$. Define*

$$\tilde{k}^{(r)}(\mathcal{M}_i, \mathcal{M}) = [\mathbf{U}_{\leq r} \mathbf{U}_{\leq r}^\top \mathbf{k}]_i \tag{20}$$

*where $\mathbf{U}_{\leq r} = (\mathbf{u}_k^\top)_{k=1}^r \in \mathbb{R}^{n \times r}$ is the sub-matrix of the orthonormal matrix $\mathbf{U} \in \mathbb{R}^{n \times n}$. Then we have*

1. *For any $r \leq n$ and $i, j$, $\tilde{k}^{(r)}(\mathcal{M}_i, \mathcal{M}_j) = K^{(r)}_{\mathcal{M}_i, \mathcal{M}_j}$ and hence $\tilde{f}^{(r)}(\mathcal{M}_i) = \hat{f}^{(r)}(\mathcal{M}_i)$ for all $i = 1, \ldots, n$.*

2. *For $r = n$ and any $i$ and any test point $\mathcal{M}$, $\tilde{k}^{(n)}(\mathcal{M}_i, \mathcal{M}) = k^{(n)}(\mathcal{M}_i, \mathcal{M})$ and hence $\tilde{f}^{(n)}(\mathcal{M}) = \hat{f}^{(n)}(\mathcal{M}) = \hat{f}(\mathcal{M})$.*

*Proof.* We use the standard notation in kernel theory: $\tilde{k}_{\boldsymbol{X}, \boldsymbol{x}_i}^{(r)} = [\tilde{k}^{(r)}(\mathcal{M}_j, \mathcal{M}_i)]_{j=1}^n$, and analogously for $k_{\boldsymbol{X}, \boldsymbol{x}_i}$. The first statement comes from:

$$\tilde{k}_{\boldsymbol{X}, \boldsymbol{x}_i}^{(r)} = \mathbf{U}_{\leq r} \mathbf{U}_{\leq r}^\top K_{\boldsymbol{X}, \boldsymbol{x}_i} = \mathbf{U}_{\leq r} \mathbf{U}_{\leq r}^\top \sum_{k=1}^n \mu_k u_{ik} \mathbf{u}_k = \sum_{k=1}^n \mu_k u_{ik} \mathbf{U}_{\leq r} \mathbf{U}_{\leq r}^\top \mathbf{u}_k = \sum_{k=1}^r \mu_k u_{ik} \mathbf{u}_k = k_{\boldsymbol{X}, \boldsymbol{x}_i}^{(r)}.$$

The second statement comes from:

$$\tilde{k}_{\boldsymbol{X}, \boldsymbol{x}}^{(n)} = \mathbf{U} \mathbf{U}^\top k_{\boldsymbol{X}, \boldsymbol{x}} = K_{\boldsymbol{X}, \boldsymbol{x}} = k_{\boldsymbol{X}, \boldsymbol{x}}^{(n)}.$$

$\square$

In short, the above theorem establishes that for $r \leq n$ (1) our approximated TKRR predictor $\tilde{f}^{(r)}$ coincides with the TKRR predictor $\hat{f}^{(r)}$ on the training set, and (2) for $r = n$ it coincides with the original KRR predictor $\hat{f}$ on any new test points. As an independent contribution, this result extends the definition of TKRR beyond the original formulation in Amini et al. (2022), which may be of interest to kernel theorists.

## THE USE OF LARGE LANGUAGE MODELS

In this work, we used large language models (LLMs) primarily as assistive tools to improve the clarity, grammar, and presentation of the manuscript. LLMs were employed to polish writing, rephrase sentences for readability, and ensure consistency in terminology. The use of LLMs did not influence the scientific content or conclusions of the paper; their role was limited to language refinement.

