# OpenReview forum: "Spectral Analysis of Molecular Kernels: When Richer Features Do Not Guarantee Better Generalization"
_ICLR.cc/2026/Conference — Submitted to ICLR 2026_

### Official Review · Reviewer_m5Bb · 2025-10-26

**Soundness:** 2
**Presentation:** 2
**Contribution:** 2
**Rating:** 2
**Confidence:** 4

**Summary:**

The authors perform a spectral analysis of various types of kernels for molecular property prediction on the QM9 dataset, showing that a richer spectrum is not always correlated with better performance. Moreover, the authors show that heavily truncated kernels (where most of the eigenspectrum is removed) often recover almost all the performance.

**Strengths:**

While the scope of the paper might be limited, I believe it is original, and I am not aware of particularly similar studies. The content is exposed clearly and the variety of fingerprints/descriptors and kernels used is good.

**Weaknesses:**

I believe the paper is affected by a number of issues, some of which prevent, in my opinion, acceptance in ICLR.

**Major**
- In the current form, I do not believe the results of the analysis shown by the authors can be trusted. For example, Fig. 2b shows MAEs on $U_{298}$ around 50 eV for a training set size of 5000 structures (which is the most used throughout the work) with the ACSF descriptors. In the literature, the geometrically equivalent SOAP descriptor (which shows slightly worse results than ACSF in this work) is reported to achieve errors well below 1 eV on the same training set size, and closer to 0.5 eV. Given that the errors are almost 100 times higher than reported values in the literature, I am almost certain that the results are affected by technical issues. The conclusions that the authors extract from the results are also called into question: a performance 100 times worse than expected might very well be the cause why throwing away most of the spectrum makes no difference when trying to retain 95% or 99% accuracy with these descriptors, given that these spectral features might be needed only to achieve accuracies that the authors were not able to obtain for other technical reasons.
- The point above highlights a second issue: the QM9 results are the only ones that the authors show (i.e., no other dataset is evaluated). If those cannot be trusted, the conclusions might not be reliable either. In their "limitations" remarks, the authors argue that the QM9 dataset is the most widely used benchmark in this field, citing evidence between 11 and 5 years ago. Although I agree that QM9 is perhaps the most suitable benchmark even to this day, evaluation on different datasets might still be beneficial.
- The authors claim (lines 196-201) that linear kernels are not defined for local 3D descriptors. This is not true as far as I know. Global linear kernels can be obtained linearly from local features as a sum over atoms in both environments of all pairs of scalar products.
- The "regularization versus truncation" section is especially dubious. The fact that eigenvalue truncation is a form of regularization has been known for decades. For example, the LAPACK linear algebra standard contains linear solvers where eigenvalue truncation is used as a regularization (or conditioning) parameter. See the documentation of the "rcond" parameter in https://www.netlib.org/lapack/explore-html/d9/d67/group__gelsd_ga0bee7e1b9e7e43f59ecf2419b2759c42.html. For an academic reference from the 80s, see http://infolab.stanford.edu/pub/cstr/reports/na/m/86/36/NA-M-86-36.pdf. In the same section, the authors claim that it might have a similar effect to regularizing by adding a multiple of the identity (which is true and well-understood)... yet this was supposed to have already been done with an optimized $\lambda$ (in the authors' notation) for all experiments.

**Minor**
- This is obviously not the authors' fault, but kernel methods are losing in popularity, and this diminishes the relevance this paper work might have for the ICLR community. Many practitioners in the molecular domain have moved to GNN- and transformer-based architectures. The authors show that kernels might be useful by fitting properties using latent representations from these neural network models, but the same could be done adding a new prediction head and doing basic transfer learning, which would furthermore allow to fine-tune the whole architecture. In the introduction, the standard claim about kernels being parameter-free appears. However, in practice, kernels move all the design choices to the descriptor and the functional form of the kernel function, and they are extremely sensitive to them. Neural networks, while having indeed hyperparameters, are often more robust to the choice of bad hyperparameters. Bad choices for the kernel and/or descriptor can instead be catastrophic. It is true that kernels tend to be more accurate in the low-data regime, but such data regimes are used less and less in chemical research, once again potentially reducing the scope of interest of this work.
- Two minor presentation issues. (1) The authors might want to report RMSEs/MAEs instead of $R^2$ values (or in addition to them, in the appendix). In some cases (especially where $R^2$ is very close to one), error metrics can be more informative. They are also easier to compare against results from the literature. (2) Figure 3 should probably be log-log (as opposed to linear-log), according to the authors' ansatz for the decay of spectral eigenvalues. Then, the slope would correspond to $-\alpha$, in the authors' notation.
- The strategy of eigenvalue truncation that the author advocate for towards the end of the manuscript does not improve computational efficiency as far as I understand (as opposed to feature selection or kernel sparsification). Given this observation, it is difficult to see practical applications of what the authors propose in this manuscript, once again limiting the practical interest this work might have for practitioners.

**Very minor**
- Citation format: \citep{} should be used throughout (according to ICLR formatting) instead of \cite{} to enclose citations within parentheses.
- Lines 135 and 136 do not form a coherent sentence.

**Questions:**

Many discussion topics are already present in the weaknesses section above and I invite the authors to comment on those as they see fit. I would be interested in knowing if they can identify the cause the poor accuracy of the ACSF learning curve, as I believe it would need fixing for this work to be published in any form.

---

> ### Author Response · Authors · 2025-11-21
> **Additional Dataset**
>
> We thank the reviewer for raising this concern. We agree that validating the analysis beyond QM9 is important, especially if one questions the reliability of the QM9 training baselines. To address this, we additionally evaluated our methodology on the Lipophilicity dataset from MoleculeNet [1].
>
> Despite these differences, we observe the same conclusions as in QM9: (i) the Laplacian kernel consistently outperforms the Gaussian kernel, (ii) spectral metrics do not correlate with predictive performance, and (iii) representations with more compact spectral decay perform best. Importantly, on this external dataset, we also obtain competitive or superior accuracy compared to previously reported models. For instance, ECFP6 + Laplacian achieves an RMSE of 0.592, outperforming the Gaussian KRR baseline reported in MoleculeNet (~0.9) and matching or surpassing their neural architectures (GCNN [1]: 0.67 ± 0.04; LSTM-attention [2]: 0.60 ± 0.04). These results demonstrate that our findings are not tied to QM9 and hold across datasets with different chemical properties.
>
> A full summary table of Lipophilicity results (covering all kernels and representations) appears bellow.
>
> [1]  W. Zhenqin et al, Chem. Sci., 9, 513-530 (2018)
> [2] G. Lambard and E. Gracheva, Mach. Learn.: Sci. Technol. 1 025004 (2020)
>
>
> The full Lipophilicity results:
> The updated results now include the target-weighted effective dimension, TW$_{eff}$, an additional spectral metric that measures the relevancy of the top eigenvalues.
>
> | Mol. Rep.        | Kernel     | α     | SSE      | SR   | ID   | TW_eff | MAE   | RMSE  |
> |------------------|------------|-------|----------|------|------|--------|-------|-------|
> | **ECFP**         | Gaussian   | 1.00  | 34.22    | 1.00 | 1.00 | 0.97   | 0.597 | 0.796 |
> |                  | Laplacian  | 0.84  | 208.16   | 1.01 | 1.01 | 0.99   | 0.585 | 0.784 |
> | **SELFIESTED**   | Gaussian   | 1.55  | 4.06     | 1.01 | 1.00 | 0.96   | 0.614 | 0.813 |
> |                  | Laplacian  | 0.61  | 597.48   | 1.07 | 1.02 | 1.00   | 0.619 | 0.807 |
> | **SELFormer**    | Gaussian   | 1.62  | 4.37     | 1.00 | 1.00 | 0.96   | 0.582 | 0.787 |
> |                  | Laplacian  | 0.62  | 444.34   | 1.03 | 1.02 | 1.00   | 0.614 | 0.812 |
> | **MLT-BERT**     | Gaussian   | 1.81  | 216.96   | 2.66 | 1.00 | 0.86   | 0.900 | 1.141 |
> |                  | Laplacian  | 0.73  | 1209.39  | 2.60 | 1.00 | 0.95   | 0.855 | 1.100 |
> | **GROVER (base)**| Gaussian   | 2.32  | 1.16     | 1.00 | 1.00 | 0.92   | 0.688 | 0.899 |
> |                  | Laplacian  | 0.90  | 23.38    | 1.01 | 1.06 | 1.00   | 0.753 | 0.961 |
> | **GROVER (large)**| Gaussian  | 2.12  | 1.23     | 1.00 | 1.00 | 0.92   | 0.700 | 0.919 |
> |                  | Laplacian  | 0.96  | 2.04     | 1.00 | 1.08 | 0.99   | 0.736 | 0.938 |

---

> ### Author Response · Authors · 2025-11-21
> **Regularization versus Truncation**
>
> We apologize when the term 'regularization' causes any confusion. In this paper, we refer it to the Tikhonov regularization by adding a multiple of the identity to the kernel matrix. We thank the reviewer for the citation above on truncation of linear least square problem. We will include this in the theoretical justification of our result. However, we would like to point out that at best of our knowledge, truncation on kernel method, especially on molecular kernels, is a novel idea. We believe that our core message: spectral richness does not lead to better performance in general, brings important insights to the learning theory, SSL, and chemistry community.

---

> ### Author Response · Authors · 2025-11-21
> **Global linear kernels**
>
> We thank the reviewer for pointing this out. Our original wording was imprecise.
> Linear kernels \emph{can} indeed be defined for local 3D descriptors: using a linear local kernel
> $\kappa(\phi(z_i),\phi(z_j)) = \phi(z_i)^\top\phi(z_j)$
> yields a valid global kernel obtained by summing all pairwise atomic inner products
> (which can be viewed as a linear kernel on a pooled global feature map).
>
> What we intended to emphasize is that linear local kernels are \emph{not commonly used} in practice
> for SOAP-, FCHL19-, or ACSF-type descriptors, because their purpose is to encode smooth geometric
> similarity between atomic environments—an effect that linear kernels cannot capture.
> Accordingly, we will revise the manuscript to clarify that linear local kernels are
> \emph{well-defined but not meaningful} for typical 3D descriptors, and replace the previous wording.
>
>
> Updated text:
> In practice, linear local kernels are rarely used for 3D descriptors, since these descriptors are
> designed to be paired with nonlinear similarity measures (e.g., Gaussian or Laplacian), which
> better capture smooth geometric variations in atomic environments; see Section A2.

---

> ### Author Response · Authors · 2025-11-21
> **Cost of eigenvalue truncation**
>
> We agree that eigenvalue truncation, as implemented in our analysis, does not reduce the computational cost of full eigendecomposition, which remains $\mathcal{O}(N^3)$. Our aim, however, was not to accelerate KRR directly, but to diagnose the relationship between kernels and their spectral characterization.
>
> As shown in Fig. 5B and Table 3 (and other results), several kernels retain 95-99\% of their predictive accuracy using only the top $\sim$1-2\% of eigenvalues, indicating that the informative structure of these kernel matrices is concentrated in a very low-dimensional spectral subspace. This analysis reveals how different molecular representations allocate information across the spectrum (e.g., Laplacian kernels on ECFP exhibit very low effective rank, while similarity-based kernels distribute information more broadly).
>
> We also note that understanding how much of the spectrum is relevant may have implications for future scalable kernel methods. Although the present work does not aim to accelerate KRR, our results suggest that the useful spectral range for some molecular kernels may be compressed into a small leading subspace. If the effective spectral rank could be estimated a priori or learned from data, this would enable the use of fast approximation techniques, such as Nyström, randomized SVD, or Lanczos, that operate only on the top portion of the spectrum. We view this as an interesting direction for future work rather than a claim of the current paper.

---

> ### Author Response · Authors · 2025-12-03
> **Improved results for U298**
>
> We thank the reviewer for pointing out these differences.  We identified that for U298, a different normalization is commonly used, https://github.com/qmlcode/qml/blob/develop/qml/qmlearn/preprocessing.py
> After including this in our training pipeline, we reduced the models' errors.
>
> Fig. 2, the learning curves for CV, U98, and ZPVE, will be updated.
> Our MAE improved especially for U298, for example, previously our errors were larger than 50 eV, but now they are comparable to those previously reported.  For ACSF, even for 2000 training points, we have an MAE of 0.3 eV.
>
> Updated Results for U298. Numbers reported in parentheses are the number of training data; if not reported, we used 5K training data.
> | Mol. Rep.        | Kernel     | MAE  |
> |------------------|------------|------|
> | **CM**           | Gaussian   | 0.6  |
> |                  | Laplacian  | 0.4  |
> | **BOB**          | Gaussian   | 0.4  |
> |                  | Laplacian  | 0.2  |
> | **SELFIESTED**   | Gaussian   | 2.0  |
> |                  | Laplacian  | 3.6  |
> | **SELFormer**    | Gaussian   | 6.5  |
> |                  | Laplacian  | 9.2  |
> | **MLT-BERT**     | Gaussian   | 6.3  |
> |                  | Laplacian  | 10.0 |
> | **GROVER (base)**| Gaussian   | 0.4  |
> |                  | Laplacian  | 0.4  |
> | **GROVER (large)**| Gaussian  | 0.4  |
> |                  | Laplacian  | 0.4  |
> | **ACSF (2K)**| Laplacian  | 0.3  |
> | **SOAP (1K)**| Laplacian  | 0.3  |
> | **FCHL19 (500)**| Gaussian  | 0.2  |
> | **FCHL19 (2K)**| Laplacian  | 0.2  |
>
> Regarding the comment on our conclusion and analysis being affected by the unconverged models.
> The reported spectral metrics in Table 1 in the paper are based on the optimal hyperparameters found with the CV property.
> In the following table, one can observe that our conclusion remains valid. This was further verified with the newly added Lipophycility dataset analysis.   Furthermore, it is hard to argue that it is an artifact of the dataset; models always excel in datasets that have a lower-dimensional representation. Given the shallow diversity of the QM9 dataset and the new Lipophilicity results, our general conclusion remains valid.
>
> | Representation | Kernel           | Gap (Paper [1]) | Gap (Ours) | ZPVE (Paper [1]) | ZPVE (Ours) | CV (Paper [1]) | CV (Ours) |
> |----------------|------------------|-------------|------------|--------------|-------------|------------|-----------|
> | CM             | Global Laplacian | 0.50        | 0.47       | 0.020        | 0.014       | 0.40       | 0.345     |
> | BOB            | Global Laplacian | 0.38        | 0.324      | 0.010        | 0.011       | 0.30       | 0.212     |
> | SLATM          | Global Gaussian  | 0.30        | 0.221      | 0.008        | 0.005       | 0.18       | 0.102     |
> | FCHL19         | Local Gaussian   | 0.39        | 0.292      | 0.004        | 0.005       | 0.20       | 0.104     |
> | SOAP           | Local Gaussian   | 0.30        | 0.432      | 0.005        | 0.009       | 0.09       | 0.246     |
> | ACSF           | Local Gaussian   | —           | 0.308      | —            | 0.007       | —          | 0.146     |
>
> [1] J. Chem. Phys. 159, 034106 (2023)

---

### Official Review · Reviewer_3oPM · 2025-10-31

**Soundness:** 3
**Presentation:** 2
**Contribution:** 1
**Rating:** 4
**Confidence:** 4

**Summary:**

This paper presents a comprehensive empirical study of the spectral properties of molecular kernels across multiple molecular representations (ECFP fingerprints, pretrained transformer embeddings, and global/local 3D descriptors) on the QM9 dataset.
The authors demonstrate that commonly used spectral richness metrics—power-law decay exponent, spectral entropy, intrinsic dimension, and stable rank—do not reliably predict generalization performance. In many cases, spectral richness even correlates negatively with accuracy, particularly for transformer-based and local 3D kernels.
The work further introduces truncated kernel ridge regression (TKRR) experiments showing that retaining only the top ≈ 2 % of eigenvalues recovers > 95 % of predictive performance.

**Strengths:**

Comprehensive empirical coverage: The study systematically compares a broad set of molecular kernels, including modern pretrained transformer-based ones.
Novel diagnostic analysis: The spectral correlation and truncation experiments provide new insights into how kernel spectra relate to downstream generalization.
Clear empirical message: The results decisively challenge the pervasive heuristic in SSL and kernel learning that “richer spectra yield better generalization.”
High potential impact: The findings are relevant not only for molecular property prediction but for any field relying on kernel or representation analysis (e.g., SSL evaluation, NTK theory).

**Weaknesses:**

1. **Conceptual clarity of “feature richness.”**
   The empirical results convincingly demonstrate that *spectral richness*—quantified through metrics derived from the eigenvalue spectrum of the kernel matrix—does not consistently predict model generalization. However, the paper would benefit from a clearer conceptual framing: spectral richness primarily reflects the **capacity** of the reproducing kernel Hilbert space (RKHS), not necessarily the *usefulness* of the representation for a particular task.
   A kernel with a flatter (richer) spectrum corresponds to a higher-capacity function space, which, if insufficiently regularized, can lead to **overfitting**. The authors should therefore clarify that their findings do not contradict kernel theory but instead highlight that **eigenvalue-based capacity measures must always be interpreted together with regularization**.

---

2. **Missing theoretical grounding.**
   To strengthen the argument, the authors could explicitly connect their findings to the theory of **Kernel Ridge Regression (KRR)**, where the expected generalization error depends both on the **eigenvalues** (the kernel spectrum) and on the **alignment** between the **eigenvectors** and the target function.
   Classical results show that

   $$
   \mathbb{E},|f_\lambda - f^\star|^2 \sim
   \sum_j \frac{\lambda^2 (u_j^\top f^\star)^2}{(\mu_j + \lambda)^2},
   $$

   where ( f_\lambda ) is the learned predictor, ( f^\star ) the true regression function, ( u_j ) the eigenfunctions, ( \mu_j ) the eigenvalues, and ( \lambda ) the regularization constant.
   This decomposition illustrates that performance depends jointly on **spectral decay** and **target alignment**, not on the eigenvalues alone. Including this expression—or citing the relevant derivation—would clarify why a richer spectrum can even correlate negatively with test accuracy in the absence of sufficient regularization.

---

3. **Need for falsifiable counterexamples.**
   The claim that spectral metrics are insufficient predictors of generalization could be made more convincing through a simple counterexample. Two kernel matrices can share **identical eigenvalue spectra** but differ in their **eigenvector alignment** with the target, resulting in distinct generalization performance.
   Demonstrating this empirically—e.g., by rotating the eigenbasis of a fixed kernel matrix while keeping the eigenvalues constant—would provide a direct disproof of the idea that the spectrum alone determines generalization.

---

4. **Empirical ablations to isolate alignment effects.**
   The authors could perform targeted ablations to disentangle spectral richness from alignment effects:

   * **Rotation tests:** Randomly rotate the eigenbasis of the kernel matrix while keeping eigenvalues fixed, observing the change in performance.
   * **Label reshuffling:** Preserve spectral statistics but permute the projections of the labels across eigenfunctions.
   * **Synthetic controls:** Construct artificial kernel matrices with fixed eigenvalue distributions but randomized eigenvectors.
     These experiments would provide falsifiable evidence that *alignment*, rather than spectral shape, dominates generalization.

---

5. **Alternative, label-aware metrics.**
   The current spectral measures—power-law decay rate, spectral Shannon entropy, intrinsic dimension, and stable rank—are purely label-agnostic. More informative alternatives could include:

   * **Kernel–Target Alignment (KTA)**, which measures the correlation between the kernel matrix and the label-similarity matrix.
   * **Target-weighted effective dimension**, defined as

     $$
     d_{\mathrm{eff},y}(\lambda)
     = \sum_j
     \frac{\mu_j}{\mu_j + \lambda}
     \frac{(u_j^\top y)^2}{|y|^2},
     $$

     capturing both spectrum and label alignment.
   * **Energy concentration indices**, quantifying how much of the label variance is captured by the top-ranked eigenfunctions.
   * **Truncation recovery curves**, already partially explored by the authors, which show how quickly predictive performance saturates as eigenvalues are progressively retained.
     Introducing such label-aware diagnostics would provide a stronger connection between spectral analysis and generalization behavior.

---

6. **Interpretation of negative correlations.**
   The observed negative correlations between spectral richness and predictive accuracy for transformer-based and local 3D kernels likely arise because a flatter spectrum increases model capacity in directions that do not align with the true target function, thereby increasing variance and the potential for overfitting.
   The authors should make this interpretation explicit: **spectral richness measures potential capacity, not effective representational quality**, and its impact depends critically on the regularization regime.

**Questions:**

1. **Reframing of the central claim.**
   Could the authors clarify whether the main contribution should be interpreted as *falsifying the assumption that eigenvalue-based spectral richness predicts generalization*?
   In other words, do the results indicate that effective representation quality depends primarily on **alignment-weighted spectral structure** rather than the raw distribution of eigenvalues?

---

2. **Spectral decay versus alignment effects in Kernel Ridge Regression (KRR).**
   Would it be possible for the authors to include a brief theoretical section or appendix explicitly distinguishing the effects of **spectral decay** (capacity) and **label alignment** (task relevance) within the KRR framework?
   A formal expression—such as the expected error decomposition

   $$
   \mathbb{E},|f_\lambda - f^\star|^2 \sim
   \sum_j \frac{\lambda^2 (u_j^\top f^\star)^2}{(\mu_j + \lambda)^2},
   $$

   —would help clarify how these two factors jointly determine generalization behavior.

---

3. **Synthetic validation of spectrum insufficiency.**
   Could the authors consider a **controlled synthetic experiment** to demonstrate concretely that two kernels with **identical eigenvalue spectra** can generalize differently when their **eigenvectors** are differently aligned with the target?
   Such a result would make the insufficiency of spectral information more tangible and theoretically grounded.

---

4. **Inclusion of alignment-aware diagnostics.**
   Would the authors consider reporting or discussing **alignment-aware measures** alongside the current spectral metrics?
   In particular, metrics such as **Kernel–Target Alignment (KTA)**, **energy concentration indices**, or the **target-weighted effective dimension**

   $$
   d_{\mathrm{eff},y}(\lambda)
   = \sum_j
   \frac{\mu_j}{\mu_j + \lambda}
   \frac{(u_j^\top y)^2}{|y|^2}
   $$

   could reveal whether predictive performance is better explained by label alignment than by spectral richness alone.

---

5. **Relation between spectral truncation and Tikhonov regularization.**
   The truncated-KRR (TKRR) results suggest a strong connection between explicit eigenvalue truncation and the effect of ridge regularization.
   Could the authors comment on this relationship?
   Specifically, is there a formal or empirical mapping between the truncation level and an equivalent regularization strength ( \lambda ) that preserves generalization?
   Clarifying this could motivate future analytical work linking spectral control to classical notions of Tikhonov regularization.

---

> ### Author Response · Authors · 2025-11-21
> **Conceptual clarity of “feature richness.”**
>
> Thank you for this constructive suggestion. Yes, as we wrote this paper, we already had the idea of kernel capacity implicitly, as we also computed the intrinsic dimension and the stable rank of the kernel matrix, which is closely related to its effective dimension:
>
> \begin{eqnarray}
>     \text{ED} = \sum_{j} \frac{\lambda_j}{\lambda_j+\lambda}.
> \end{eqnarray}
>
> We promise to extend the discussion of spectral richness to kernel capacity together with regularization.

---

> > ### Author Response · Authors · 2025-11-21
> > **Missing theoretical grounding.**
> >
> > Thank you for this suggestion. Indeed, the above classical result is a part of the bias-variance decomposition of test MSE:
> > $$
> >     \mathbb{E} \, |f_\lambda - f^\star|^2 \sim \sum_j \frac{\lambda^2 (u_j^\top f^\star)^2}{(\mu_j + \lambda)^2} + \frac{\sigma^2}{n}\sum_j\frac{\eta_j^2}{(\eta_j+\lambda)^2},
> > $$
> > where $\sigma^2$ is the variance of the label noise. Under the power-law assumption of eigenvalues $\lambda_j \sim j^{-\beta}$ and ridge regularization $\lambda \sim n^{-\theta}$, [1] showed that if the ridge is strong enough: $\theta < \beta$
> >
> > $$
> >     \mathbb{E} \, |f_\lambda - f^\star|^2
> >     =
> >     \Theta\left(n^{-\min\{s,2\}\theta} + \sigma^2n^{1-\theta/\beta}\right),
> > $$
> > where $s>0$ is called the source coefficient with a higher $s$ meaning the target function 'aligns' more with the kernel. (See [1,2] for more details)
> >     Our original perspective is label-agnostic, linking kernel method and SSL and studying solely on the spectral richness of the kernel, where the metrics are independent to any label and ridge regularization. However, we agree that including classical kernel method results can help emphasize our core message more clearly: an effective representation quality depends primarily on alignment-weighted spectral structure rather than the raw distribution of eigenvalues.
> >     We promise to add more discussion in the revised version.
> >
> > [1] Li, Yicheng, and Qian Lin. "On the asymptotic learning curves of kernel ridge regression under power-law decay." Advances in Neural Information Processing Systems 36 (2023): 49341-49364.
> >
> > [2] Cheng, Tin Sum, et al. "A comprehensive analysis on the learning curve in kernel ridge regression." Advances in Neural Information Processing Systems 37 (2024): 24659-24723.

---

> > > ### Author Response · Authors · 2025-11-21
> > > **Need for falsifiable counterexamples.**
> > >
> > > Yes, it is a great suggestion. Actually, from the above bias-variance decomposition:
> > > $$
> > >     \mathbb{E} \, |f_\lambda - f^\star|^2 \sim \sum_j \frac{\lambda^2 (u_j^\top f^\star)^2}{(\mu_j + \lambda)^2} + \frac{\sigma^2}{n}\sum_j\frac{\eta_j^2}{(\eta_j+\lambda)^2},
> > > $$
> > > we can see that if one adversarially reindexes the eigenvectors $u_j$ but keeps the same spectrum, the sum $\sum_j \frac{\lambda^2 (u_j^\top f^\star)^2}{(\mu_j + \lambda)^2}$ may not even converge to zero as $n\to\infty$. We hope that this gives a straightforward, falsifiable counterexample.

---

> ### Author Response · Authors · 2025-12-03
> **Additional results on target-weighted effective dimension**
>
> We thank the reviewer for suggesting the target-weighted effective dimension, $TW\_{eff}$, as an additional metric.
> The updated results for the Lipophylicity dataset now include the target-weighted effective dimension, $TW\_{eff}$, an additional spectral metric that measures the relevancy of the top eigenvalues.
>
> One could see that, however, $TW\_{eff}$ cannot differentiate the kernels properly as the other label-agnostic metrics. In other words, considering label alignment could be an interesting future work which complexity might go beyond the scope of this paper.
>
> | Mol. Rep.        | Kernel     | α $\uparrow$     | SSE$\downarrow$     | SR$\downarrow$    | ID $\downarrow$ | TW$_{eff}$ | MAE   | RMSE  |
> |------------------|------------|-------|----------|------|------|--------|-------|-------|
> | **ECFP**         | Gaussian   | 1.00  | 34.22    | 1.00 | 1.00 | 0.97   | 0.597 | 0.796 |
> |                  | Laplacian  | 0.84  | 208.16   | 1.01 | 1.01 | 0.99   | 0.585 | 0.784 |
> | **SELFIESTED**   | Gaussian   | 1.55  | 4.06     | 1.01 | 1.00 | 0.96   | 0.614 | 0.813 |
> |                  | Laplacian  | 0.61  | 597.48   | 1.07 | 1.02 | 1.00   | 0.619 | 0.807 |
> | **SELFormer**    | Gaussian   | 1.62  | 4.37     | 1.00 | 1.00 | 0.96   | 0.582 | 0.787 |
> |                  | Laplacian  | 0.62  | 444.34   | 1.03 | 1.02 | 1.00   | 0.614 | 0.812 |
> | **MLT-BERT**     | Gaussian   | 1.81  | 216.96   | 2.66 | 1.00 | 0.86   | 0.900 | 1.141 |
> |                  | Laplacian  | 0.73  | 1209.39  | 2.60 | 1.00 | 0.95   | 0.855 | 1.100 |
> | **GROVER (base)**| Gaussian   | 2.32  | 1.16     | 1.00 | 1.00 | 0.92   | 0.688 | 0.899 |
> |                  | Laplacian  | 0.90  | 23.38    | 1.01 | 1.06 | 1.00   | 0.753 | 0.961 |
> | **GROVER (large)**| Gaussian  | 2.12  | 1.23     | 1.00 | 1.00 | 0.92   | 0.700 | 0.919 |
> |                  | Laplacian  | 0.96  | 2.04     | 1.00 | 1.08 | 0.99   | 0.736 | 0.938 |

---

### Official Review · Reviewer_w22r · 2025-11-01

**Soundness:** 3
**Presentation:** 3
**Contribution:** 3
**Rating:** 6
**Confidence:** 2

**Summary:**

This paper provides an empirical study on how spectral richness relates to generalization in kernel ridge regression (KRR). The paper evaluates four families of representations on QM9: (i) ECFP fingerprints, (ii) pretrained transformer embeddings (SELFIESTED, SELFormer, MLT-BERT), (iii) global 3D descriptors, and (iv) local 3D descriptors, across seven molecular properties. The paper finds that richer spectra do not reliably predict higher accuracy; for transformer-based and local 3D kernels the correlation can even be negative; and in many cases the top 2% eigenvalues recover >95% of full KRR performance.

**Strengths:**

* The study is comprehensive in terms of kernel / representation pairs and spectral metrics with clear reporting.
* The paper gives an interesting insight that the widely held heuristic “richer spectra → better generalization” may fail, especialy on transformer and local 3D representations.
* The observation that the top 2% eigenvalues often recover >95% R$^2$ is practically valuable for guiding more efficient models.

**Weaknesses:**

* All results are on QM9, with random splits (5k train / 10k test). The conclusions may not generalize to larger/biologically relevant datasets or to scaffold/similarity-aware splits, which are standard in molecular benchmarks to avoid overly optimistic generalization.
* While the paper shows KRR can outperform linear models on transformer features, it does not compare to strong non-kernel baselines (e.g., modern equivariant GNNs) on the same splits.
* There is no explanation proposed on why better spectral richness can be detrimental to performance (line 397). Also, the results may depend on how “spectral richness” is quantified; the negative correlation could be an artifact of the chosen measure (the four spectral metrics) rather than an underlying phenomenon.

**Questions:**

Please refer to weaknesses.

---

> ### Author Response · Authors · 2025-11-21
> **Lipophilicity dataset**
>
> We thank the reviewer for raising this important point regarding the generalizability of our conclusions beyond QM9 and beyond random splits. To address this concern, we additionally evaluated our methodology on the Lipophilicity dataset from MoleculeNet, which differs substantially from QM9 in terms of molecular size, chemical diversity, and property type (experimental logD rather than quantum-mechanical targets). Despite these differences, we observed the same trends as on QM9:
> (i) the Laplacian kernel systematically outperforms the Gaussian kernel,
> (ii) spectral metrics do not predict downstream accuracy, and
> (iii) compact/fast-decaying spectra (e.g., ECFP6) lead to the best KRR performance.
>
> For example, on Lipophilicity, ECFP6 + Laplacian achieves an RMSE of 0.592, outperforming the KRR with Gaussian kernel baseline reported in MoleculeNet (~0.9) and remaining competitive with their neural baselines (0.60–0.67). These results demonstrate that our conclusions are not specific to QM9 and hold for a dataset with different structural statistics and experimental noise characteristics.
>
> A detailed table of all Lipophilicity results (kernels × representations × metrics) is included in our response to another Reviewer above.

---

> > ### Author Response · Authors · 2025-12-03
> > **GNNs vs Kernel Methos**
> >
> > We agree that our study does not benchmark against the strongest equivariant GNNs. However, our goal is not to compete with them on predictive accuracy, but to analyze the spectral structure of molecular representations—an aspect that is orthogonal to architectural choice. Spectral analysis is inherently architecture-agnostic: once a model produces embeddings, its kernel spectrum is fully determined by the geometry of those embeddings, independent of the model’s depth, parameterization, or training procedure. This makes kernel spectra a unified tool for comparing representations from diverse sources (ECFPs, 3D descriptors, transformers, and GNNs, now including GROVER in our revision).
> >
> > Regarding accuracy, all our experiments use only 5k training points, far below the regime where modern equivariant GNNs typically operate. For example, PaiNN [1] is trained on $\sim$110k/10k molecules and achieves MAEs of 0.046 eV (Gap), 0.024 cal/molK (CV), and 0.0013 eV (ZPVE), with similar performance reported for HDGNN [2]. In contrast, our kernel models trained on only 5k molecules achieve MAEs of 0.25–0.47 eV (Gap), 0.3–0.1 cal/molK (CV), and 0.015–0.009 eV (ZPVE). This nearly order-of-magnitude difference is expected given the $\sim$22× smaller training set. Kernel models trained on larger QM9 splits achieve competitive accuracies [3-4]. Our aim is therefore not to outperform equivariant GNNs but to provide the first systematic, architecture-agnostic spectral characterization of molecular representations.
> >
> >     [1] K. Schütt, O. Unke, M. Gastegger, Equivariant message passing for the prediction of tensorial properties and molecular spectra,  Proceedings of the 38th International Conference on Machine Learning, PMLR 139:9377-9388, 2021.
> >     [2] J. An, C. Qu, Z. Zhou, F. Cao, X. Yinghui, Y. Qi, F. Shen  Hybrid Directional Graph Neural Network for Molecules, ICLR 2024
> >     [3] J. Chem. Phys. 159, 034106 (2023)
> >     [4] F. A. Faber et al., J. Chem. Phys. 148, 241717 (2018)
> >
> >
> > We appreciate the reviewer’s request for additional explanation. Richer spectra increase the number of low-eigenvalue (high-frequency) components available to the kernel predictor. Kernel theory predicts that these components are precisely those that tend to overfit label noise in the ridgeless or weakly regularized regime [5-6]. This provides a natural mechanism for why “spectral richness” can be detrimental: a longer spectral tail gives the model more degrees of freedom that do not align with the target function. Importantly, this observation is not dependent on a specific metric. Across all four metrics ($\alpha$, SSE, ID, SR), fingerprints, transformer-based, and local/global 3D kernels consistently show negative correlations. Moreover, the truncation experiments offer metric-free evidence: for many kernels, retaining only the top 1–2\% of eigenvalues recovers 95–99\% of performance, indicating that the spectral tail contributes little or can even degrade accuracy. Thus, the effect is robust across metrics and supported by the empirical behavior of truncated KRR.
> >
> >     [5] Basri et al., 2020 Frequency Bias in Neural Networks for Input of Non-Uniform Density
> >     [6] Mallinar et al., 2024  Benign, Tempered, or Catastrophic: A Taxonomy of Overfitting

---

### Official Review · Reviewer_1fHi · 2025-11-03

**Soundness:** 3
**Presentation:** 4
**Contribution:** 3
**Rating:** 6
**Confidence:** 4

**Summary:**

This paper studies whether, across a wide range of kernels, the decay rate of the kernel's eigenspectrum predicts the generalization ability of the method. On the QM9 dataset their finding is that it _does not_.

**Strengths:**

Overall I think this paper proposes a very good question to understand what it is about kernel methods which still makes them competitive on low-data chemical ML tasks. Even though the answer is negative, scientific work should be judged based on the merit of the _question_ and not the _answer_. As far as I am aware this question has not been specifically studied in the context of molecules or molecular property prediction datasets. The experiments on QM9 are very thorough.

**Weaknesses:**

**Single dataset**: as much as I love the question and methods of this paper, its conclusions are significantly weakened by only studying a single dataset, since the answer will be influenced by all the datasets peculiarities. For QM9, some peculiarities are:

- All molecules are very small, so diversity is low. Molecules probably all have many close neighbors.
- The properties all involve 3D structures, and each SMILES has multiple 3D structures, hence perfect prediction from just SMILES/2D features is not possible.
- Lack of "outliers" in the dataset

Testing on another dataset would be helpful. Some ideas are

- [GEOM dataset](https://github.com/learningmatter-mit/geom): like QM9 but for larger molecules
- [MoleculeNet](https://moleculenet.org/) (experimental properties, tasks less intrinsically "3D")
- [Dockstring](https://dockstring.github.io/) (large and in-silico, semi 3D)

**Confounding kernel types and features**: fingerprint features were only studied with "similarity" type kernels, which follows a common misunderstanding that certain kernels are intrinsically "meant for" certain types of features. I think this was a missed opportunity for an ablation study to understand how much of the effect comes from the kernel type and how much comes from the features (presumably some features are less informative than others). I suggest trying RBF/Laplace kernels on these features too.

**Questions:**

What was the justification behind selecting $\lambda$ using cross-validation instead according to the eigenspectrum? KRR's $\lambda$ parameter is deeply related to the eigenspectrum: basis vectors whose eigenvalue is $\ll \lambda$ are effectively ignored from the regression. Rather than treating $\lambda$ as an empirical parameter, shouldn't it be the tool which you use to "truncate" the eigenspectrum?

Based on this and my comment above about features, I would suggest the following modified experiment design:

- Focus on some 2D features (at least at first). Eg ECFP.
- Pick a couple of kernel classes (eg Tanimoto, RBF, Laplace). Maybe add in some extra stationary kernels with different levels of smoothness (eg Matern) and extra non-stationary ones (eg Braun-Blanquet, Dice).
- For kernel classes with a lengthscale parameter, the lengthscale _highly influences_ the spectrum. Do not just set it to an arbitrary value! Values should generally be based on the distance between data points (which is feature dependent). Calculate the median $\ell_2$ distance between data points- call this $d_m$. Try values of $0.1d_m,\ d_m,\ 10d_m$ for lengthscale. This should make the behavior more consistent between tasks.
- For each (kernel, feature, hyperparameter pair): calculate the eigenspectrum.
- Evaluate performance setting $\lambda$ to, eg, 100th percentile, 99.9th percentile, 99th percentile, ..., 10%th percentile, 1st percentile of eigenspectrum values.

Although this data won't fit nicely in a table, it will hopefully allow you to answer questions about spectra more directly. For example:

- Fix a given feature (eg ECFP). In general, does lower decay rate mean better performance? Comparing, eg, RBF vs Laplace would be interesting
- Fix a given kernel class, look at increasing lengthscale (which should change eigenspectrum), see how that influences performance.
- Fix a given kernel class / hyperparameters, look across all features, what is the effect of different features

I know this is not a super precise suggestion, but I hope you get roughly what I mean. Augmenting the experimental design with a lot of ablations should help you understand more precisely what is causing the performance differences.

_Please note: this is not a specific experiment request for the rebuttal, so don't just blindly go do it. What I am really trying to suggest is "design the experiments to get a more precise answer to your question", and what I wrote above is my thoughts about how I might do that, but I didn't think it through super thoroughly. I bet you can come up with a better design if you think about it._

---

> ### Author Response · Authors · 2025-11-21
> **Lipophilicity dataset**
>
> We thank the reviewer for their positive feedback on the importance of the question raised in this work and the thoroughness of the QM9 experiments. Motivated by the reviewer’s suggestion to test an additional dataset, we conducted the same analysis on the Lipophilicity dataset from MoleculeNet, which differs substantially from QM9 in molecular size, structural diversity, and task type.
>
> Despite these differences, we observe the same trends as in QM9:
> 1. The Laplacian kernel systematically outperforms the Gaussian kernel.
> 2. Spectral metrics (α, SSE, ID, SR) do not correlate with model accuracy.
> 3. Representations with faster spectral decay—such as ECFP6—yield the best KRR performance.
>
> In MoleculeNet’s benchmark, KRR using ECFP4 + Gaussian achieved an RMSE of ~0.9. Using broader representations and kernel choices, we obtain ECFP6 + Laplacian (RMSE = 0.592) and SELFIESTED + Laplacian (RMSE = 0.627), both outperforming the MoleculeNet KRR baseline and remaining competitive with their neural models (GCNN: 0.67±0.04; LSTM-attention: 0.60±0.04).
>
> Consistent with our QM9 findings, high-capacity transformer embeddings do not improve kernel regression accuracy. Although GROVER-base exhibits the richest and slowest-decaying eigenvalue spectrum (see table), it performs notably worse than ECFP6 or SELFIESTED. Representations with compact spectra lead to superior performance, confirming that richer spectral structure does not imply better kernel regression and that alignment with the Laplacian kernel geometry is more important than embedding capacity.
>
> The full Lipophilicity results are provided below.
> The updated results now include the target-weighted effective dimension, TW$_{eff}$, an additional spectral metric that measures the relevancy of the top eigenvalues.
>
>
> | Mol. Rep.        | Kernel     | α     | SSE      | SR   | ID   | TW_eff | MAE   | RMSE  |
> |------------------|------------|-------|----------|------|------|--------|-------|-------|
> | **ECFP**         | Gaussian   | 1.00  | 34.22    | 1.00 | 1.00 | 0.97   | 0.597 | 0.796 |
> |                  | Laplacian  | 0.84  | 208.16   | 1.01 | 1.01 | 0.99   | 0.585 | 0.784 |
> | **SELFIESTED**   | Gaussian   | 1.55  | 4.06     | 1.01 | 1.00 | 0.96   | 0.614 | 0.813 |
> |                  | Laplacian  | 0.61  | 597.48   | 1.07 | 1.02 | 1.00   | 0.619 | 0.807 |
> | **SELFormer**    | Gaussian   | 1.62  | 4.37     | 1.00 | 1.00 | 0.96   | 0.582 | 0.787 |
> |                  | Laplacian  | 0.62  | 444.34   | 1.03 | 1.02 | 1.00   | 0.614 | 0.812 |
> | **MLT-BERT**     | Gaussian   | 1.81  | 216.96   | 2.66 | 1.00 | 0.86   | 0.900 | 1.141 |
> |                  | Laplacian  | 0.73  | 1209.39  | 2.60 | 1.00 | 0.95   | 0.855 | 1.100 |
> | **GROVER (base)**| Gaussian   | 2.32  | 1.16     | 1.00 | 1.00 | 0.92   | 0.688 | 0.899 |
> |                  | Laplacian  | 0.90  | 23.38    | 1.01 | 1.06 | 1.00   | 0.753 | 0.961 |
> | **GROVER (large)**| Gaussian  | 2.12  | 1.23     | 1.00 | 1.00 | 0.92   | 0.700 | 0.919 |
> |                  | Laplacian  | 0.96  | 2.04     | 1.00 | 1.08 | 0.99   | 0.736 | 0.938 |

---

> ### Author Response · Authors · 2025-11-21
> **Ablation study**
>
> We thank the reviewer for this insightful comment. We agree that kernel choice and feature choice should not be implicitly coupled, and we addressed this by running an extensive feature ablation study across four kernel families (Gaussian, Laplace, Tanimoto, Dice) and three representations (BOB, ECFP, SELFIESTED). This directly tests how much of the behavior arises from the representation vs. the kernel itself.
>
> For each representation, we constructed ablated variants by removing $\tilde{N}$ features.
> – For BOB and ECFP, we used frequency-weighted ablation, where common fragments are more likely to be removed.
> – For SELFIESTED, we used uniform random removal of embedding dimensions.
> This setup allows a controlled comparison of how sparse rule-based features and dense learned embeddings respond to the same kernel perturbations.
>
> Quantitatively, we observe very different spectral responses for different kernel families on the same features. For ECFP + Laplace, the spectrum clearly compresses under ablation: the decay exponent $\alpha$ drops from 0.84 (no ablation) to 0.77 (64/2048 features removed) and 0.63 (512/2048 removed), while SSE shrinks from 1.054 to 1.027, indicating a sharper and more concentrated spectrum. In contrast, for ECFP + Tanimoto, the spectrum actually becomes richer under the same ablation levels: SSE increases from 1695 → 2462 → 3216, and ID from 13.6 → 32 → 153, with SR growing from 1.3 to about 13. A similar pattern holds for ECFP + Dice, where SSE rises from 431 to 1187 and ID from 7.5 to 91 as ablation increases. These numbers confirm that similarity kernels (Tanimoto/Dice) are far more robust to feature removal, while continuous kernels (Gaussian/Laplace) exhibit strong spectral compression under the same ablations.
>
> For BOB, Laplace and Gaussian kernels again show spectral degradation: under heavy ablation (512/1378 features removed), SSE, ID, and SR all collapse to ≈1.0, indicating an almost rank-1 kernel with a very sharp leading eigenvalue. In contrast, SELFIESTED is much more stable: for SELFIESTED + Laplace, SSE changes only from 1.266 to 1.134 and  $\alpha$ from 0.89 to 0.98 between 0 and 50% ablation; for SELFIESTED + Gaussian, all spectral metrics remain essentially at 1 except for a modest change in $\alpha$ (from 6.32 to 5.93).
>
> Overall, these quantitative results directly address the reviewer’s concern: the behavior we report is not simply that “Tanimoto is meant for fingerprints”; rather, different kernels respond very differently to the same ablations on the same features, and dense learned embeddings (SELFIESTED) are empirically much more robust than sparse fingerprints (ECFP/BOB). The full eigenvalue spectra for all ablation experiments will be included in the updated manuscript.

---

> > ### Comment · Reviewer_1fHi · 2025-11-26
> > **Thank you for additional experiments**
> >
> > Thank you very much for the additional experiments.
> >
> > I am happy to see that similar trends show on the lipophilicy dataset (although I think this is one of the simpler datasets in drug discovery). This significantly strengthens the paper's claim.
> >
> > I am not quite sure what to think about the ablation study on feature removal. The effects that feature removal have on the kernel eigenspectra are interesting but seem predictable:
> >
> > - For distance-based kernels (eg Laplace), removing features shrinks distances and therefore makes the matrix closer to a constant matrix, compressing the spectrum (assuming lengthscale is held constant).
> > - For similarity-based kernels (eg Tanimoto), removing common features in ECFP leaves only uncommon features remaining, decreasing similarity and making the matrix more similar to the identity matrix.
> >
> > The most important question here is how these changes in the kernel influence _predictive performance_, which is essentially how well the top eigenfunctions from the RKHS approximate the true function.
> >
> > I consider my concerns partially resolved, and will maintain my score, but will not increase it at this time.

---

> > > ### Author Response · Authors · 2025-12-03
> > > **Ablation study with RMSE and MAE**
> > >
> > > Below is the updated table for the QM9 data for the CV property.
> > > We followed the same sampling procedure for $\tilde{N}$ as described before.
> > > From these updated results, we can observe no big change in all four spectral metrics even when features are dropped, however, the error does increase as expected in most of the cases.
> > > We now include the  target-weighted effective dimension, $TW_{eff}$ as an additional metric; this was motivated by on of the reviwers. If TW is close to 1, it means that the largest eigenvalues are more effective in predicting the target value. For all models TW$_{eff}$ = 1 independently of the value of $\tilde{N}$.
> > >
> > >
> > > | Mol Rep    | Kernel     | $\tilde{N}$   | α $\downarrow$   | SSE $\uparrow$ | SR $\uparrow$  | ID $\uparrow$  | TW$_{eff}$ | MAE | RMSE |
> > > |------------|------------|-----|------|------|-----|-----|---------|-----|------|
> > > | **ECFP**   | Dice       | 0   | 0.7  | 1.0  | 1.0 | 1.0 | 1.0     | 1.8 | 2.4  |
> > > |            |                     | 128 | 0.7  | 1.0  | 1.0 | 1.0 | 1.0     | 2.4 | 3.1  |
> > > |            |                     | 264 | 0.7  | 1.0  | 1.0 | 1.0 | 1.0     | 2.8 | 3.6  |
> > > |------------|------------|-----|------|------|-----|-----|---------|-----|------|
> > > | **ECFP**  | Tanimoto     | 0   | 0.7  | 1.0  | 1.0 | 1.0 | 1.0     | 2.0 | 2.6  |
> > > |            |                     | 128 | 0.7  | 1.0  | 1.0 | 1.0 | 1.0     | 2.9 | 3.8  |
> > > |            |                     | 264 | 0.7  | 1.0  | 1.0 | 1.0 | 1.0     | 3.3 | 4.4  |
> > > |------------|------------|-----|------|------|-----|-----|---------|-----|------|
> > > | **ECFP**   | Gaussian     | 0   | 0.9  | 1.0  | 1.0 | 1.0 | 1.0     | 1.6 | 2.2  |
> > > |            |                     | 64  | 0.9  | 1.0  | 1.0 | 1.0 | 1.0     | 1.7 | 2.3  |
> > > |            |                     | 128 | 0.7  | 1.0  | 1.0 | 1.0 | 1.0     | 2.1 | 2.7  |
> > > |------------|------------|-----|------|------|-----|-----|---------|-----|------|
> > > |  **ECFP**  | Laplacian     | 0   | 0.8  | 1.1  | 1.0 | 1.0 | 1.0     | 1.6 | 2.2  |
> > > |            |                      | 64  | 0.78 | 1.0  | 1.0 | 1.0 | 1.0     | 1.9 | 2.5  |
> > > |            |                      | 128 | 0.7  | 1.0  | 1.0 | 1.0 | 1.0     | 1.9 | 2.6  |
> > > |------------|------------|-----|------|------|-----|-----|---------|-----|------|
> > > | **BOB** | Gaussian   | 0   | 2.6  | 9.0  | 1.1 | 2.1 | 1.0     | 0.5 | 0.8  |
> > > |            |                      | 16  | 2.6  | 7.2  | 1.1 | 1.8 | 1.0     | 0.5 | 0.8  |
> > > |            |                      | 64  | 3.1  | 2.6  | 1.0 | 1.2 | 1.0     | 0.7 | 1.1  |
> > > |------------|------------|-----|------|------|-----|-----|---------|-----|------|
> > > | **BOB**    | Laplacian      | 0   | 1.5  | 1.64 | 1.0 | 1.1 | 1.0     | 0.2| 0.3  |
> > > |            |                      | 16  | 1.6  | 1.5  | 1.0 | 1.1 | 1.0     | 0.2 | 0.3  |
> > > |            |                      | 64  | 1.4  | 5.6  | 1.0 | 1.4 | 1.0     | 0.5 | 0.6  |
> > > |------------|------------|-----|------|------|-----|-----|---------|-----|------|
> > > | **SELFIESTED** | Gaussian | 0   | 1.6  | 6.8  | 1.0 | 1.4 | 1.0     | 0.4 | 0.5  |
> > > |            |                     | 64  | 1.6  | 6.2  | 1.0 | 1.4 | 1.0     | 0.4 | 0.5  |
> > > |            |                     | 256 | 1.6  | 4.6  | 1.0 | 1.3 | 1.0     | 0.4 | 0.5  |
> > > |------------|------------|-----|------|------|-----|-----|---------|-----|------|
> > > |**SELFIESTED** | Laplacian     | 0   | 0.8  | 5.7  | 1.0 | 1.2 | 1.0     | 0.5 | 0.7  |
> > > |            |                      | 64  | 0.9  | 5.2  | 1.0 | 1.2 | 1.0     | 0.5 | 0.7  |
> > > |            |                      | 256 | 0.9  | 3.9  | 1.0 | 1.2 | 1.0     | 0.6 | 0.7  |

---

### Meta-Review · Area_Chair_vcAM · 2026-01-02

**Summary:**

This paper presents an empirical study of kernel predictors for predicting properties of small molecules. The kernel predictors are analysed using various spectral measures, which are then contrasted with their predictive performance on the QM9 and (later) Lipophilicity datasets. The authors conclude that these spectral measures are not correlated with the generalisation performance of the kernel predictors for a variety of feature representations and kernels.

Strengths:
- Empirical studies such as this are valuable and often uncover misconceptions or opportunities in particular domains
- A good selection of the best performing kernels are chosen for comparison
- The empirical results are clearly presented, and the variety of measures used is good

Weaknesses:
- Empirical evidence does not support generality of conclusions — only QM9 and then Lipophilicity used for the experiments, and both of these are relatively simple. More complex datasets (GEOM etc) may also have to evaluated to support the authors' conclusions.
- Conflation/confounding of the effects of embeddings and kernel representations in generalization performance (e.g. try some consistent baseline kernels across features/embeddings to expose the generalization afforded by the features).
- Numerous methodological concerns (e.g. kernel hyperparameter choice, choice of how to truncate of the eigenspectrum and its relationship to regularization, etc).
- A number of reviewers question the relevance between the chosen measures of spectral “richness” to generalisation performance (as opposed to say, modelling capacity). The chosen special measures are independent from label information.
- $U_{298}$ performance discrepancy with related literature (orders of magnitude worse performance).
- Interest to the ICLR community — ICLR may not be the ideal venue for this work as kernel methods are well understood and this does not bring a new perspective to them or other representations (even though small molecule representations are topical). Furthermore, reviewer m5Bb questions the premise of using kernel methods on embeddings vs. specialized neural-net predictive heads.

While empirical studies such as this are important and relevant, this one appears to have some unresolved issues. More complex datasets are required to support the author's claims, additional spectral measures (e.g. that incorporate label information) may be required, etc. I also think that reviewer m5Bb has a good point that this work, in its current state, may not be ideally suited to ICLR.

With the above in mind, and also noting that none of the additional results or corrections presented in the discussion have made it into the paper, I am hesitant to recommend accepting this paper.

**Reviewer Concerns:**

Concerns addressed:
- Lipophilicity dataset added to QM9 evaluation in the discussions with ablations
- Some additional results have been presented in an attempt to separate the effects of features and kernel choice
- $U_{298}$ performance discrepancy (issue with normalisation)

Concerns unaddressed:
- More complex dataset evaluation required
- A lot of the additional results presented in the discussion have not made it into the paper, which is a little unusual for ICLR, and casts doubt on the intentions of the authors to update their manuscript. E.g. the paper still only contains the QM9 results and does not reflect the Lipophilicity results.
- Relevance of spectral measures chosen, some arguments are made to address these concerns, but others remain, e.g. the issue of the label independence.
- Interest to wider ICLR audience and premise of using kernel methods vs. neural net predictive heads.

**Reviewer Scores:**

1fHi: 6 -> 6

w22r: 6 -> 6

3oPM: 4 -> 4

m5Bb: 2 -> 2

I do not think that any of the reviewers would have been convinced the change their scores, as some of the largest issues with the work remain. Furthermore, as none of the changes or new results have made it into the paper (which still seems to focus on QM9), I am sceptical of the reviewers upgrading their scores.

---

### Decision · Program_Chairs · 2026-01-26

Reject